# Continuous-Multiple Image Outpainting in One-Step via Positional Query and A Diffusion-based Approach

**Shaofeng Zhang**[1], **Jinfa Huang**[2], **Qiang Zhou**[3], **Zhibin Wang**[3], **Fan Wang**[4], **Jiebo Luo**[2], **Junchi Yan**[1]*

[1]Department of CSE & MoE Key Lab of Artificial Intelligence, Shanghai Jiao Tong University
[2]University of Rochester, [3]INF Tech Co., Ltd., [4]Alibaba Group
`{sherrylone, yanjunchi}@sjtu.edu.cn`
`{jhuang90@ur,jluo@cs}.rochester.edu`
Code: https://github.com/Sherrylone/PQDiff

## Abstract

Image outpainting aims to generate the content of an input sub-image beyond its original boundaries. It is an important task in content generation yet remains an open problem for generative models. This paper pushes the technical frontier of image outpainting in two directions that have not been resolved in literature: 1) outpainting with arbitrary and continuous multiples (without restriction), and 2) outpainting in a single step (even for large expansion multiples). Moreover, we develop a method that does not depend on a pre-trained backbone network, which is in contrast commonly required by the previous SOTA outpainting methods. The arbitrary multiple outpainting is achieved by utilizing randomly cropped views from the same image during training to capture arbitrary relative positional information. Specifically, by feeding one view and positional embeddings as queries, we can reconstruct another view. At inference, we generate images with arbitrary expansion multiples by inputting an anchor image and its corresponding positional embeddings. The one-step outpainting ability here is particularly noteworthy in contrast to previous methods that need to be performed for $N$ times to obtain a final multiple which is $N$ times of its basic and fixed multiple. We evaluate the proposed approach (called PQDiff as we adopt a diffusion-based generator as our embodiment, under our proposed **P**ositional **Q**uery scheme) on public benchmarks, demonstrating its superior performance over state-of-the-art approaches. Specifically, PQDiff achieves state-of-the-art FID scores on the Scenery (**21.512**), Building Facades (**25.310**), and WikiArts (**36.212**) datasets. Furthermore, under the 2.25x, 5x and 11.7x outpainting settings, PQDiff only takes **40.6%**, **20.3%** and **10.2%** of the time of the benchmark state-of-the-art (SOTA) method.

## 1 Introduction

Image outpainting (Lin et al., 2021a; Cheng et al., 2022; Wang et al., 2021), a.k.a. image extrapolation (Wang et al., 2022; Kim et al., 2021; Zhang et al., 2020), is to generate new content beyond the original boundaries of a given sub-image. It is technically an essential problem for generative models and remains relatively open (compared with other condition-based generation settings e.g. image inpainting (Bertalmio et al., 2000), style transfer (Luan et al., 2017) etc.), which meanwhile can find wide applications in automatic creative image, virtual reality. Usually, an ideal outpainter is expected to achieve the following basic functions (Yao et al., 2022): 1) determining where the missing regions should be located relative to the output's spatial locations for both nearby and faraway features; 2) guaranteeing that the extrapolated image has a consistent structural layout with the given sub-image; and 3) the borders between extrapolated regions and original input images should be visually smooth.

---

*Junchi Yan is the correspondence author. The SJTU authors are partly supported by NSFC (62222607) and Shanghai Municipal Science and Technology Major Project (2021SHZDZX0102) and SJTU Trans-med Awards Research (STAR) 20210106.

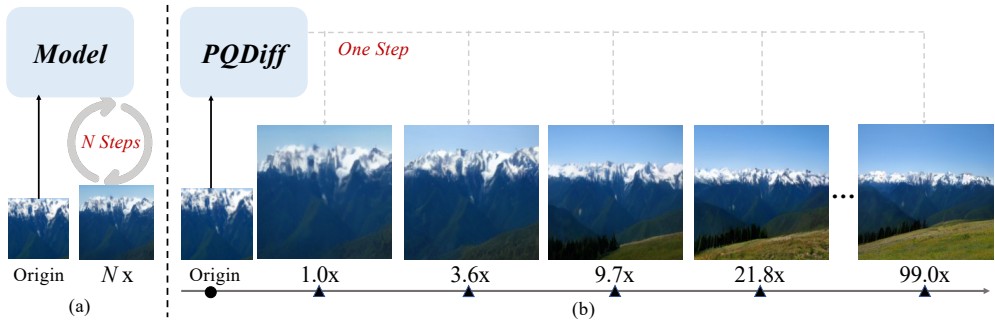

Figure 1: PQDiff can outpaint images with arbitrary and continuous multiples in one step (b). In contrast, previous methods (a) outpaint images with discrete multiples in multiple steps. Note that $N$x here means an $N$-times larger image needs to be generated, while 2.25x, 5x, and 11.7x are adopted in the experiment following the setting of the previous work (Yao et al., 2022) for fair comparisons.

Table 1: Methodology comparisons of the proposed PQDiff with recent advanced image outpainting methods. Pixel loss means pixel-wise $l_1$ or $l_2$ loss on the generated images. Feature loss means $l_2$ loss on the feature map. No pretrain means the backbone is randomly initialized.

| Method | Generation type | Objectives | Encoder | Continuous | One-step | No pretrain |
|---|---|---|---|---|---|---|
| NSIPO (Yang et al., 2019) | GAN-based | Pixel+GAN loss | ResNet-50 | ✗ | ✗ | ✔ |
| IOH (Van Hoorick, 2019) | GAN-based | Pixel+GAN loss | Conv layers | ✗ | ✗ | ✔ |
| Uformer (Gao et al., 2023) | GAN-based | Pixel+GAN+Feature Loss | Swin Trans | ✗ | ✗ | ✗ |
| QueryOTR (Gao et al., 2023) | GAN+MAE | Pixel+GAN Loss | ViT | ✗ | ✗ | ✗ |
| Vanilla Diff (Ho et al., 2020) | Diffusion-based | Pixel Loss | ViT | ✗ | ✔ | ✔ |
| PQDiff | Diffusion-based | Pixel Loss | ViT | ✔ | ✔ | ✔ |

These requirements have been mainly addressed by existing outpainting methods which in general fall into two categories: i) GAN-based methods (Van Hoorick, 2019; Yang et al., 2019), whereby random noise and the initial input sub-images (as conditions) are used to generate the fake surrounding image content, and the discriminator is to classify the generated images as fake or real; ii) MAE-based methods (Yao et al., 2022) use MAE (Masked Autoencoder) (He et al., 2022) as the main architecture. They model the extrapolation as the MIM (Masked Image Modeling) problem (Xie et al., 2022) by replacing the extrapolated regions around the input sub-images with masked tokens and predicting the pixels of the masked patches. Specifically, these MAE-based methods also employ a discriminator to enhance the smoothness of the borders between extrapolated regions and the original sub-images. The above two kinds of methods mainly suffer from two applicability limitations. **First,** as shown in Fig. 1(a), they require running multiple times to outpaints the image (e.g., the SOTA method in (Yao et al., 2022) outpaints 11.7x images by passing through the model three times (x $\rightarrow forward \rightarrow$ 2.25x $\rightarrow forward \rightarrow$ 5x $\rightarrow forward \rightarrow$ 11.7x), which is inefficient especially when the required expansion multiple is large (e.g. 99x); **Second,** the discriminator-based architecture can slow down the convergence speed (Goodfellow et al., 2014; Adler & Lunz, 2018), and require a pre-trained encoder. In other words, the computational cost of training their model in fact includes the pretraining cost, which is usually very high (e.g. up to 1,000 or longer epochs on ImageNet).

In this paper, we emphasize and tackle two less-studied challenges, especially the flexibility, and efficiency for a practical tool: the model is expected to i) outpaint images in arbitrary and continuous multiples (i.e. by $N \in \mathbb{R}$ times), where $N$x multiples mean the contents of the extrapolated images are $N > 1$ times larger than the input sub-images, especially without resorting to retrain a model for every different $N$; and ii) outpaint with any multiple $N$ in one step[1].

Recently diffusion models (Dhariwal & Nichol, 2021; Pokle et al., 2022; Jin et al., 2023b;c; Augustin et al., 2022) have shown success for multi-modal (Jin et al., 2023a; 2022), segmentation (Tan et al., 2022; Wu et al., 2021), backbone designing (Wu et al., 2022; Dai et al., 2022), and image generation (Saharia et al., 2022) with a progressive denoising procedure. Yet such an iterative step-by-step (also called timesteps) in the sampling (i.e. testing) stage can be too tedious for outpainting, especially considering image outpainting itself also still requires its own iterations

---

[1]Note that the step here refers to how many iterations are needed to be passed through the model instead of **timesteps** in the diffusion models.

(e.g. QueryOTR (Yao et al., 2022), IOH (Van Hoorick, 2019)). To achieve one-step diffusion-based generation, we propose to use relative positional queries and input sub-images as conditions. Since the relative positional embedding can represent any positional relationship between the input sub-image and the extrapolated image, we can outpaint the sub-image in controllable and continuous multiples in one step (Fig. 1(b)). We make methodology comparisons in Table 1 to better position our method. The main contributions include:

**i) Continuous multiples for image outpainting.** We propose PQDiff, which learns the positional relationships and pixel information at the same time. Specifically, in the training stage, PQDiff first randomly crops the given images twice to generate two views. Then, PQDiff learns one cropped view from the other cropped view and the pre-calculated relative positional embeddings (RPE) of the two views. Since the RPE can represent continuous relationships between two views, PQDiff can outpaint the images in continuous multiples. To our best knowledge, we are the first to outpaint images in continuous multiples (e.g., 1x, 2.25x, 3.6x, 21.8x), whereas the SOTA QueryOTR (Yao et al., 2022) can only outpaint images in discrete multiples.

**ii) One-step image outpainting.** We propose a position-aware cross-attention mechanism between relative positional embedding and input sub-image patches, which helps PQDiff to outpaint images in only one step for any multiple settings. As far as we know, PQDiff is the first to achieve this capability, whereas (Yao et al., 2022; Yang et al., 2019) can only outpaint images step-by-step, which severely limits their sampling, i.e. generation efficiency. Under the 2.25x, 5x and 11.7x outpainting settings, PQDiff only takes **40.6%**, **20.3%**, and **10.2%** of the time of QueryOTR (Yao et al., 2022).

**iii) New SOTA performance.** Experimental results on outpainting benchmarks (Gao et al., 2023; Yang et al., 2019) show that PQDiff significantly surpasses QueryOTR (Yao et al., 2022) and achieves new SOTA **21.512**, **25.310** and **36.212** FID scores with the challenging 11.7x multiple setting on the Scenery, Building Facades, and WikiArts datasets, respectively. Moverover, PQDiff achieves new SOTA results in most settings (2.25x, 5x, and 11.7x).

## 2 BACKGROUND AND RELATED WORK

**Image Outpainting.** It aims to generate the surrounding regions from the visual content, which can be considered as an image-conditioned generation task (Odena et al., 2017; Kang et al., 2021; Guo et al., 2020; Arjovsky et al., 2017; Gulrajani et al., 2017). The work (Sabini & Rusak, 2018) brings the image outpainting task to attention with a deep neural network inspired by image inpainting (Bertalmio et al., 2000). It focuses on enhancing the quality of generated images smoothly by using GANs and post-processing to perform horizontal outpainting. The work (Van Hoorick, 2019) designs a CNN-based encoder-to-decoder framework by using GAN for image outpainting. In (Wang et al., 2019), a Semantic Regeneration Network is proposed to directly learn the semantic features from the conditional sub-image. While a 3-stage model is developed in (Lin et al., 2021b) with an edge-guided generative network to produce semantically consistent output. Although these methods avoid bias in the general padding and up-sampling pattern, they still suffer from blunt structures and abrupt color issues, which tend to ignore spatial and semantic consistency. To tackle these issues, a Recurrent Content Transfer (RCT) block is devised (Yang et al., 2019) for temporal content prediction with Long Short Term Memory (LSTM) networks (Hochreiter & Schmidhuber, 1997). To enrich the context, (Lu et al., 2021) additionally switches the outer area of images into its inner area.

**The SOTA Outpainting Method QueryOTR (Yao et al., 2022).** We particularly discuss QueryOTR for its so-far best performance as well as its adoption of ViT module as also will be used in our approach. QueryOTR (Yao et al., 2022) proposes to adopt the ViT-based encoder and MIM-based (Zhang et al., 2023c) architecture for outpainting. Specifically, given an input sub-image $\mathbf{x} \in \mathbb{R}^{H \times W \times 3}$, QueryOTR first partitions $\mathbf{x}$ into regular non-overlapping patches with the patch size $P \times P$ to obtain the patch tokens $\{\mathbf{x}_p^1, \mathbf{x}_p^2, \cdots, \mathbf{x}_p^L\}$, where $P$ is usually set as 16 and $L = \frac{H \times W}{P^2}$. The goal of QueryOTR is to predict the extra sequence $\{\mathbf{x}_p^{L+1}, \mathbf{x}_p^{L+2}, \cdots, \mathbf{x}_p^{L+R}\}$ representing the extrapolated regions. In line with MAE (He et al., 2022), the sin-cos positional embedding of input patches is pre-defined. Thus, the positional embedding of the extrapolated patches can be obtained. The training of QueryOTR also follows MAE (He et al., 2022), where the visible patches sequence $\{\mathbf{x}_p^1, \mathbf{x}_p^2, \cdots, \mathbf{x}_p^L\}$ is fed to the encoder. Then, the fixed positional embedding of the masked tokens sequences $\{\mathbf{x}_p^{L+1}, \mathbf{x}_p^{L+2}, \cdots, \mathbf{x}_p^{L+R}\}$ is used as input to the decoder to predict the extrapolated

patches. Finally, QueryOTR copies the input sub-image to the generated image in the corresponding position, followed by the Patch Smoothing Module (PSM) to smooth the border.

**Diffusion Models.** These models (e.g. the seminal work (Ho et al., 2020)) gradually inject noise into data and then reverse this process to generate data from noise. The noise-injection process is also called the forward process. Given the original data $\mathbf{x}_0$ (clean), then, the forward process can be formalized as a Markov chain: $q(\mathbf{x}_{1:T}|\mathbf{x}_0) = \prod_{t=1}^{T} q(\mathbf{x}_t|\mathbf{x}_{t-1})$ where $q$ is the forward process and $q(\mathbf{x}_t|\mathbf{x}_{t-1}) = \mathcal{N}(\mathbf{x}_t|\sqrt{\alpha_t}\mathbf{x}_{t-1}, \beta_t\mathbf{I})$, and $\alpha$ and $\beta$ represent the noise schedule and $\alpha + \beta = 1$. $\mathcal{N}(0,1)$ means the standard Gaussion noise. To reverse this process, a Gaussion model $p(\mathbf{x}_{t-1}|\mathbf{x}_t) = \mathcal{N}(x_{t-1}|\mu_t(\mathbf{x}_t), \sigma_t^2\mathbf{I})$ is adopted to approximate the ground truth reverse transition $q_{\mathbf{x}_{t-1}|\mathbf{x}_t}$. Specifically, the optimal mean value of $\mathbf{x}_t$ can be written as (Bao et al., 2022): $\mu_t^*(\overline{\alpha_t}) = \frac{1}{\sqrt{\alpha_t}}\left(\mathbf{x}_t - \frac{\beta_t}{\sqrt{1-\overline{\alpha}}}\mathbb{E}[\epsilon|\mathbf{x}_t]\right)$ where $\overline{\alpha_t} = \prod_{i=1}^{t} \alpha_i$, and $\epsilon$ is the standard Gaussian noises injected to $\mathbf{x}_t$. Thus, the learning is equivalent to a noise prediction task. Formally, a noise prediction network $\epsilon_\theta(\mathbf{x}_t, t)$ is used to learn $\mathbb{E}[\epsilon|\mathbf{x}_t]$ by minimizing the noise prediction objective. For $l_2$ loss, we can formulate the objective of noise prediction task as $min_\theta \mathbb{E}_{t,\mathbf{x}_0,\epsilon}\|\epsilon - \epsilon_\theta(\mathbf{x}_t, t)\|_2^2$, where $t$ is uniform between 1 and $T$. On the basis of the plain diffusion models, LDM (Rombach et al., 2022) proposes to add noise and denoise in the latent space, which greatly improves the training efficiency. Followed by LDM, ViTDiff (Bao et al., 2023) proposes to replace CNN-base U-net (Ronneberger et al., 2015) with ViTs (Dosovitskiy et al., 2021) to estimate the backward process in diffusion models.

## 3 THE PROPOSED POSITIONAL-QUERY BASED DIFFUSION MODEL

We provide a concrete embodiment based on the diffusion model and we term our approach as PQDiff, whereby the continuous multiples and one-step generation are achieved. In fact, our PQ framework with these two advantages can also incorporate other generative models e.g. GANs, and the empirical performance comparison given in our ablation studies. We also show the significant improvements of our approach compared with a vanilla diffusion model directly for outpainting.

**Approach Overview.** Our approach mainly consists of key modules: relative positional embedding, diffusion process, cross attention in the position-aware transformer model, and sampling pipeline.

**Relative Positional Embedding.** Given the image $\mathbf{x} \in \mathbb{R}^{H \times W \times 3}$ from the training set, we first randomly crop the image twice and resize the cropped image to generate two views $\mathbf{x}_a \in \mathbb{R}^{h_1 \times w_1 \times 3}$ and $\mathbf{x}_b \in \mathbb{R}^{h_2 \times w_2 \times 3}$, where $\mathbf{x}_a$ and $\mathbf{x}_b$ are denoted as the anchor and target view which are not necessarily the same size. We also denote $(w, h)$ as the predefined resolution to be generated. Then, we resize both the anchor view and target view to $(w, h)$. As illustrated in Fig. 2, we can first obtain the prior information of positional relationship $(m, n)$ between the anchor view $\mathbf{x}_a$ and target view $\mathbf{x}_b$. Then, we design the relative positional embedding to represent the position relation:

$$\mathbf{E}_{m,n} = \left[\sin\left(\frac{m}{e^{2*1/d}}\right), \cos\left(\frac{m}{e^{2*2/d}}\right), \cdots, \sin\left(\frac{m}{e}\right), \ \sin\left(\frac{n}{e^{2*1/d}}\right), \cos\left(\frac{n}{e^{2*2/d}}\right), \cdots, \sin\left(\frac{n}{e}\right)\right],$$
(1)

where $e = 10,000$ is the pre-defined parameter, as also commonly used in (He et al., 2022; Zhang et al., 2023a;b). $(m, n)$ means the position of top-left patch position (see Fig. 2). Note that we randomly crop two views, and as a result, the positional relationship of the two views might be either containing, overlapping, or non-overlapping, and our model can jointly learn these three relationships.

**Forward and Backward of Conditioned Diffusion.** Given the anchor view $\mathbf{x}_a \in \mathbb{R}^{h \times w \times 3}$, target view $\mathbf{x}_b \in \mathbb{R}^{h \times w \times 3}$ and the relative positional embedding $\mathbf{E} \in \mathbb{R}^{L \times D}$ ($L$ and $D$ are the patches' length and the predefined dimension), we first encode the two views by VQVAE (Van Den Oord et al., 2017) (frozen) to compress the images to latent space, resulting in $\mathbf{z}_a \in \mathbb{R}^{h' \times w' \times c}$ and $\mathbf{z}_b \in \mathbb{R}^{h' \times w' \times c}$ (usually, $h' < h$, $w' < w$). The compression aims to improve the training efficiency and convergence speed of the diffusion models (Bao et al., 2023; Rombach et al., 2022). After obtaining the two latent views, we patchify the two views and obtain the anchor sequence $\{\mathbf{z}_a^1, \mathbf{z}_a^2, \cdots, \mathbf{z}_a^L\}$ and target sequence $\{\mathbf{z}_b^1, \mathbf{z}_b^2, \cdots, \mathbf{z}_b^L\}$ (typically, we set $L = \frac{h \times w}{p^2}$). Then, the forward process of diffusion on the target sequence can be formulated as:

$$q(\mathbf{z}_{b_t}|\mathbf{z}_{b_{t-1}}) = \mathcal{N}(\mathbf{z}_{b_{t-1}}|\sqrt{\alpha_t}\mathbf{z}_{b_{t-1}}, \beta_t\mathbf{I}), \text{ and } q(\mathbf{z}_{b_t}|\mathbf{z}_{b_0}) = \mathcal{N}(\mathbf{z}_{b_t}|\sqrt{\overline{\alpha_t}}, (1 - \overline{\alpha}_t)\mathbf{I}), \quad (2)$$

where $\mathbf{z}_{b_0}$ is the original target sequence $\{\mathbf{z}_b^1, \mathbf{z}_b^2, \cdots, \mathbf{z}_b^L\}$ and $\overline{\alpha}_t = \prod_{t=1}^{t} \alpha_i$. For the backward process, suppose we have a neural network $g_\theta$ (will be described later), taking the noisy target $\mathbf{z}_{b_t}$,

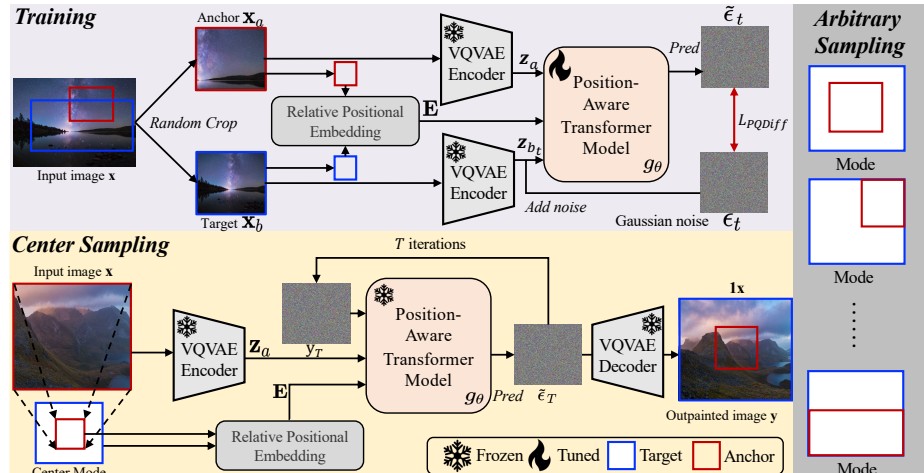

Figure 2: Framework of PQDiff. RPE in Eq. 1 means relative positional embeddings (we give the pseudo-code to calculate the RPE in Appendix A). For training, we randomly crop the image twice with different random crop ratios to obtain two views. Then, we compute the relative positional embeddings of the anchor view (red box) and the target view (blue box). For sampling, i.e. testing or generation, we first compute the target view (blue box) based on the anchor view (red box) to form a mode that means a positional relation. With different types of modes, we can perform arbitrary and controllable image outpainting. Then, we feed the RPE, random Gaussian noise, and input sub-image to perform outpainting. In theory, our PQDiff can outpaint (predict) the region at any location, due to the randomness of cropping in the training stage. We illustrate how to calculate the relative position in Appendix C. Mode means the positional relations between the anchor view and the target view.

clean anchor sequence $\mathbf{z}_a$ and relative positional embeddings $\mathbf{E}$ as input. Then, the network aims to predict the added noise $\epsilon_t$ on the target sequence $\mathbf{x}_{b_t}$. Then, the objective of PQDiff can be written as:

$$\mathcal{L}_{PQDiff} = \|\widetilde{\epsilon}_t - \epsilon_t\|_p^p, \quad and \quad \widetilde{\epsilon}_t = g_\theta(\mathbf{x}_a, \mathbf{x}_{b_t}, \mathbf{E}, t). \tag{3}$$

We set $p = 2$ in line with the previous generative methods (Rombach et al., 2022; Bao et al., 2023).

**The Position-Aware Transformer Model** $g_\theta$. Here, we describe the architecture of the neural network used in the diffusion model in detail. Consider we have the noisy target $\mathbf{z}_{b_t}$, clean anchor sequence $\mathbf{z}_a$, relative positional embeddings $\mathbf{E}$ and the timestep $t$, we first concatenate the noisy target and the anchor sequence at the channel dimension, followed by a linear layer to map to original dimension to reduce the computational cost, and we denote the mapped embedding as $\mathbf{z}_g \in \mathbf{L} \times \mathbf{D}$, where $D$ is the predefined hidden dimension in transformer network. Then, we feed the $\mathbf{z}_g$ into the transformer encoder, which is composed of several transformer blocks (Vaswani et al., 2017). After the transformer encoder, the position-aware cross-attention mechanism is proposed to learn positional relationship, which can be formulated as:

$$\mathbf{z}_d = \text{Attn}\left(\mathbf{Q_E}, \mathbf{K_{z_g}}, \mathbf{V_{z_g}}\right) = \text{Softmax}\left(\frac{\mathbf{Q_E}\mathbf{K_{z_g}}^\top}{\sqrt{D}}\right)\mathbf{V_{z_g}}, \tag{4}$$

where $\mathbf{Q}_E = \mathbf{E}\mathbf{W}_q$, $\mathbf{K}_{z_g} = \mathbf{z}_g\mathbf{W}_k$, $\mathbf{V}_{z_g} = \mathbf{z}_g\mathbf{W}_v$, and $\mathbf{W}_q, \mathbf{W}_k, \mathbf{W}_v$ are learnable parameters. After capturing the information of the target position $\mathbf{z}_d$, we directly feed the $\mathbf{z}_d$ into the transformer decoder composed of several transformer blocks, followed by a convolutional layer to predict noise.

**Sampling Pipeline.** After training the network well, we can outpaint the image in any controlled multiples, since the designed relative positional encoding can represent any positional relationship between two images. In the sampling stage, we can simply take the input sub-image as the anchor view, and input any position we want. Then, we calculate the positional encoding of the given position and feed the RPE to the network. Then, the network can predict the noise as mentioned in Eq. 3. Finally, through Eq. 2, we can simply compute the fake $\widetilde{\mathbf{z}}_{b_0}$, and predict $\mathbf{z}_{b_{t-1}}$ step-by-step by:

$$q(\mathbf{z}_{b_{t-1}}|\mathbf{z}_{b_t}, \widetilde{\mathbf{z}}_{b_0}) = \mathcal{N}(\mathbf{z}_{b_{t-1}}; \widetilde{\mu}_t(\mathbf{z}_{b_t}, \widetilde{\mathbf{z}}_{b_0}), \widetilde{\beta}_t\mathbf{I}),$$

$$\widetilde{\mu}_t(\mathbf{z}_{b_t}, \widetilde{\mathbf{z}}_{b_0}) = \frac{\sqrt{\overline{\alpha}_{t-1}}\beta_t}{1 - \overline{\alpha}_t}\widetilde{\mathbf{z}}_{b_0} + \frac{\sqrt{\alpha}(1 - \overline{\alpha}_{t-1})}{1 - \overline{\alpha}_t}\mathbf{z}_{\mathbf{b_t}}, \quad and \quad \widetilde{\beta}_t = \frac{1 - \overline{\alpha}_{t-1}}{1 - \overline{\alpha}_t}\beta_t. \tag{5}$$

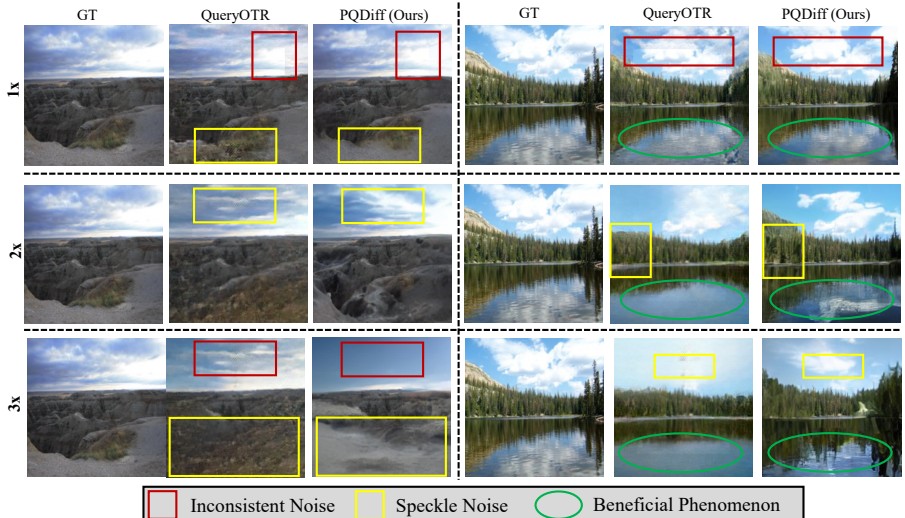

Figure 3: Comparison on the 2.25x, 5x, and 11.7x settings with the SOTA method QueryOTR. The images generated by QueryOTR come from the pre-trained model in their official repository. We highlight two kinds of noises from QueryOTR. The **red** box indicates that the boundary of the input sub-image is inconsistent with the generated region, and a **yellow** box contains noise and spots. We also find some interesting phenomena (highlighted in the **green** ovals of generated images in the right figure), where PQDiff can notice the generated "clouds", and reflect the "clouds" in "water". Moreover, the shape of the clouds in the sky and the reflections in the water are also consistent. In contrast, the previous method only generates "clouds", but ignores the reflection in the water.

After several iterations, when $t = 0$, we can obtain the extrapolated images. The training and sampling algorithms are given in Alg. 1 and Alg. 2 in Appendix A, respectively.

**Discussion.** As DDPM (Ho et al., 2020) requires step-by-step sampling, DDIM (Song et al., 2021a) is proposed to use ODE (Song et al., 2021b) equation for faster sampling. Our PQDiff can also use the DDIM for faster sampling. Specifically, through the Euler method and probability flow ODE proposed in (Song et al., 2021b), we can obtain $\mathbf{z}_{b_{t-\Delta t}}$ by:

$$\mathbf{z}_{b_{t-\Delta t}} = \sqrt{\alpha_{t-\Delta t}} \cdot \left( \frac{\mathbf{z}_{b_t}}{\alpha_t} + \frac{1}{2} \left( \frac{1 - \alpha_{t-\Delta t}}{\alpha_{t-\Delta t}} - \frac{1 - \alpha_t}{\alpha_t} \right) \cdot \sqrt{\frac{\alpha_t}{1 - \alpha_t}} \cdot g_\theta(\mathbf{z}_{b_t}) \right). \tag{6}$$

Then, the one-timestep sampling in each iteration and be replaced with $\Delta_t$ timesteps in each iteration.

## 4 EXPERIMENTS

### 4.1 EXPERIMENTAL RESULTS

**Quantitative Results.** Table 2 shows that PQDiff with a copy operation outperforms in all metrics on 2.25x, 5x, and 11.7x experiments. In particular, with a larger outpainting multiple, PQDiff can obtain better FID and IS scores. For 11.7x outpainting, PQDiff surpasses the previous SOTA QueryOTR (Yao et al., 2022) **16.460**, **25.310**, and **36.216** FID scores on Scenery, Building and WikiArt datasets, respectively. We also find an interesting phenomenon that, on WikiArt, PQDiff can surpass PQDiff + Copy on 5x and 11.7x experiments. In other words, the generated images without copy could be more realistic than the ones with the centroid copy operation. This is perhaps because the boundary region of the input sub-image is slightly inconsistent with the generated image.

**Qualitative Results.** Examples of visual results are shown in Fig. 3 (the "copy" operation is added on all methods for better comparisons). PQDiff effectively outpaints the images by querying the global semantic-similar image patches. As seen from the 2.25x outpainting results, PQDiff could generate more realistic images with vivid details and enrich the contents of the generated regions. In addition, although QueryOTR (Yao et al., 2022) adds the smoothing module to handle the inconsistency of the boundary region, there are still a few noisy spots as highlighted in **red** boxes. For 5x and 11.7x results, we can clearly see the images generated from the QueryOTR are much vaguer than those generated by PQDiff. Moreover, PQDiff can handle details well. For example, the "clouds' reflection"

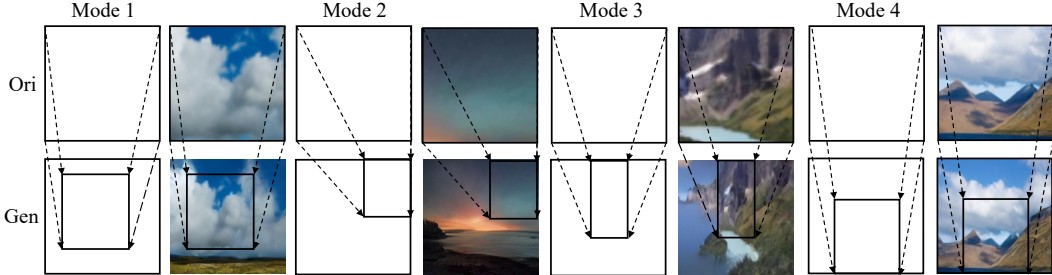

Figure 4: Example images generated by PQDiff via random relative positional embedding. The original image (Ori) as input is mapped to the corresponding location in the generated image (Gen). Note that the generated images do not explicitly undergo any "copy" operation, but still reserve the input pixels. Moreover, PQDiff learns the scales of input images according to the given mode setting.

Table 2: Results at different multiples. 2.25x, 5x, and 11.7x mean input sizes are 128x128, 86x86, and 56x56, respectively. After feeding to the network $g_\theta$, all input sub-images are first resized to 192x192. "+ Copy" operation means after the network yields the extrapolated images, the model copies the input sub-images to the corresponding center region in the extrapolated images. The best results and second best results are indicated by **boldface** and underline, respectively.

| Multiple | Methods | Type | Encoder | Scenery FID ↓ | Scenery IS ↑ | Building Facades FID ↓ | Building Facades IS ↑ | WikiArt FID ↓ | WikiArt IS ↑ |
|---|---|---|---|---|---|---|---|---|---|
| 2.25x | SRN (Wang et al., 2019) | GAN-based | VGG (Simonyan & Zisserman, 2014) | 47.781 | 2.981 | 38.644 | 3.862 | 76.749 | 3.629 |
| | NSIPO (Yang et al., 2019) | GAN-based | ResNet (He et al., 2016) | 25.977 | 3.059 | 30.465 | 4.153 | 22.242 | 5.600 |
| | IOH (Van Hoorick, 2019) | GAN-based | Conv layers | 32.107 | 2.886 | 49.481 | 3.924 | 40.184 | 4.835 |
| | Uformer (Gao et al., 2023) | GAN-based | Swin Transformer (Liu et al., 2021) | 20.575 | 3.249 | 30.542 | 4.189 | 15.904 | 6.567 |
| | QueryOTR (Yao et al., 2022) + Copy | GAN+MAE | ViT (Dosovitskiy et al., 2021) | 20.366 | 3.955 | 22.378 | 4.978 | 14.955 | 7.896 |
| | PQDiff (Ours) | Diffusion-based | ViT (Dosovitskiy et al., 2021) | 29.446 | 3.849 | 28.855 | 4.879 | 10.454 | 7.374 |
| | PQDiff + Copy (Ours) | Diffusion-based | ViT (Dosovitskiy et al., 2021) | **20.100** | **3.981** | **19.133** | **5.350** | **7.968** | **8.605** |
| 5x | SRN (Wang et al., 2019) | GAN-based | VGG (Simonyan & Zisserman, 2014) | 83.772 | 2.349 | 74.304 | 3.651 | 137.997 | 3.039 |
| | NSIPO (Yang et al., 2019) | GAN-based | ResNet (He et al., 2016) | 45.989 | 2.606 | 58.341 | 3.669 | 51.668 | 4.591 |
| | IOH (Van Hoorick, 2019) | GAN-based | Conv layers | 44.742 | 2.655 | 76.476 | 3.456 | 75.070 | 4.289 |
| | Uformer (Gao et al., 2023) | GAN-based | Swin Transformer (Liu et al., 2021) | 39.801 | 2.920 | 63.915 | 3.798 | 41.107 | 5.900 |
| | QueryOTR (Yao et al., 2022) + Copy | GAN+MAE | ViT (Dosovitskiy et al., 2021) | 39.237 | 3.431 | 41.273 | 4.547 | 43.757 | 6.341 |
| | PQDiff (Ours) | Diffusion-based | ViT (Dosovitskiy et al., 2021) | 34.492 | 3.547 | 34.799 | 4.433 | 15.297 | 6.971 |
| | PQDiff + Copy (Ours) | Diffusion-based | ViT (Dosovitskiy et al., 2021) | **28.668** | **3.712** | **29.396** | **4.763** | 15.772 | **7.876** |
| 11.7x | SRN (Wang et al., 2019) | GAN-based | VGG (Simonyan & Zisserman, 2014) | 115.193 | 2.087 | 110.036 | 2.938 | 181.533 | 2.504 |
| | NSIPO (Yang et al., 2019) | GAN-based | ResNet (He et al., 2016) | 64.457 | 2.405 | 81.301 | 3.431 | 75.785 | 4.225 |
| | IOH (Van Hoorick, 2019) | GAN-based | Conv layers | 58.629 | 2.432 | 95.068 | 2.790 | 108.328 | 3.728 |
| | Uformer (Gao et al., 2023) | GAN-based | Swin Transformer (Liu et al., 2021) | 60.497 | 2.638 | 93.888 | 3.388 | 72.923 | 5.904 |
| | QueryOTR (Yao et al., 2022) + Copy | GAN+MAE | ViT (Dosovitskiy et al., 2021) | 60.977 | 3.114 | 64.926 | 4.612 | 69.951 | 5.683 |
| | PQDiff (Ours) | Diffusion-based | ViT (Dosovitskiy et al., 2021) | 44.517 | 3.269 | 43.971 | 4.620 | **33.735** | 6.957 |
| | PQDiff + Copy (Ours) | Diffusion-based | ViT (Dosovitskiy et al., 2021) | **39.465** | **3.574** | **39.616** | **4.754** | 36.326 | **7.724** |

Table 3: Comparison of sampling time, FID, and inception scores with 2.25x, 5x, 11.7x settings on Scenery dataset. The sampling time is the wall-clock time of generating 64 images on a single GPU.

| Method | 2.25x Time (Sec.) ↓ | 2.25x FID ↓ | 2.25x IS ↑ | 5x Time (Sec.) ↓ | 5x FID ↓ | 5x IS ↑ | 11.7x Time (Sec.) ↓ | 11.7x FID ↓ | 11.7x IS ↑ |
|---|---|---|---|---|---|---|---|---|---|
| QueryOTR (Yao et al., 2022) | 23.728 | **20.366** | 3.955 | 47.456 | 39.237 | 3.431 | 71.184 | 60.977 | 3.114 |
| PQDiff (20 timesteps) | **9.638** | 20.593 | **4.026** | **9.638** | **28.138** | **3.707** | **9.638** | **42.348** | **3.737** |

in the "water" is consistent with the generated "clouds" in the sky, as highlighted in the **green** ovals. We also provide more qualitative comparisons and generated images in Appendix E.

**Sampling (i.e. Generation) Speed.** We also compare the sampling speed of PQDiff with different timesteps. Specifically, we first train PQDiff with 80,000 iterations. Then, for the sampling stage, we evaluate the pre-trained PQDiff with different timesteps. Table 3 reports the wall-clock time spent on generating 64 images on 8 V100 GPUs. Since PQDiff can outpaint images with any multiples in one step, the cases for 2.25x, 5x, and 11.7x spend almost the same time. In contrast, previous methods will take much more time under the 5x and 11.7x settings than in 2.25x. It is worth noting that, under the 2.25x setting, the inception score of PQDiff with 200 timesteps (4.111) is even higher than the ground truth (4.091) (refer to Appendix E).

## 4.2 ABLATION STUDIES

**Outpainting in an arbitrary position.** Previous outpainting methods mainly plot the same multiples around the top, down, left, and right regions, and they can simply find where the input sub-image should locate in the generated image. Thus, the "copy" operation can be simply finished. However, for outpainting in random positions, it is difficult to find the corresponding locations, since the

Table 4: Comparisons of the center PSNR score over 2.25x, 5x, and 11.7x. Note that we find PQDiff calculates several infinite values, which indicates the center parts in images generated by PQDiff are completely the same with the input sub-images without any bias. Hence, to make our model comparable, we only include the images whose PSNR scores are below 1,000. For QueryOTR, we do not find the infinite value phenomenon, and we directly report the averaged PSNR scores.

| Method | Scenery | | | Building | | | WikiArt | | |
|---|---|---|---|---|---|---|---|---|---|
| | 2.25x | 5x | 11.7x | 2.25x | 5x | 11.7x | 2.25x | 5x | 11.7x |
| QueryOTR (Yao et al., 2022) | 22.146 | 18.926 | 15.375 | 18.591 | 15.318 | 13.119 | 19.726 | 15.874 | 14.016 |
| PQDiff (Ours) | **27.676** | **27.267** | **24.697** | **28.831** | **28.614** | **27.219** | **25.946** | **25.673** | **23.372** |

two images (generated and original) could have different scales. Moreover, the two images may also not intersect. Hence, we directly illustrate the images generated by PQDiff without the "copy" operation. Some examples generated from the controlled position are shown in Fig. 4, where the images generated by PQDiff without the "copy" iteration can be still vivid and realistic. Furthermore, without the "copy" operation, the generated images can also record the pixel information and put it into the corresponding locations. Meanwhile, since the scales of input sub-images and generated images may be different, PQDiff implicitly learns to scale the input sub-images as well.

**Diversity of generated images and PSNR in center regions.** We also show five generated images with fixed positions and input in Fig. 11 in Appendix E, showing PQDiff can generate diverse content in the generated regions. Furthermore, it also retains the input pixels in the center parts of generated samples. Recall that in Sec. B, we choose not to use the PSNR score as evaluation metrics, as we also need to account for the diversity. Here we provide further analysis with PSNR score whose definition is as follows, and a higher score suggests a smaller mean square error:

$$PSNR(\mathbf{x}, \mathbf{y}) = 10 \cdot \log_{10} \left( \frac{MAX_{\mathbf{x}}^2}{\mathbb{E}_{i,j}[\mathbf{x}(i,j) - \mathbf{y}_{i,j}]^2} \right), \quad 0 \leq i, j \leq H, W \tag{7}$$

where $\mathbf{x}, \mathbf{y}$ are input images, and $H, W$ are the height and width of the input images, and $MAX_{\mathbf{x}}^2$ is a constant. We then report the PSNR score of the generated center region and input images in Table 4. Note that since the objective of QueryOTR is only added to the generated position, the center part is not regularized in the training stage. Hence, the generated center content is completely black area. Hence, for better comparison, we modify QueryOTR, adding the objective to the whole generated images with the $\mathcal{L}_{rec}$ loss. The PSNR scores in Table 4 and the generated samples in Fig. 11 demonstrate our model can: 1) memorize the input pixels and put them into corresponding locations; 2) learn the semantic information of input images, and outpaint the image with consistent contents around the images. It is worth noting that the central regions in images generated by our PQDiff without the "copy" operation can be completely the same with input images ($\mathbb{E}_{i,j}[\mathbf{x}(i,j) - \mathbf{y}_{i,j}]^2 = 0$), which indicates, our PQDiff can reconstruct the images without any bias in some cases. Since these cases without bias would lead to infinite values of the PSNR score, which will impact the average values, we directly remove these infinite values. Experimental results show even if we remove these infinite values, our method can still produce much higher center PSNR scores than QueryOTR.

**Impact of random crop ratio.** For better consistency of inputs in the training stage and sampling stage, we crop the view $\mathbf{x}_b$ with a larger crop ratio than view $\mathbf{x}_a$ (since the outpainted images are usually larger than input images). We conduct experiments to analyze the effect of random crop ratios. Specifically, we fix the crop ratio of target view $\mathbf{x}_b$ as $(0.8, 1.0)$, and switch the crop ratio of anchor views $\mathbf{x}_a$ from 0.15~0.50. We train the model with 80,000 iterations on the Scenery dataset and show the results in Fig. 5. The results show the query crop ratio influences 2.25x, 5x, and 11.7x experiments in different manners. Specifically, for 2.25x experiments, since the input images are 128x128, and the outpainted images are 192x192, where the extrapolated image is only a little larger than the input sub-image. Hence, PQDiff with larger crop ratios outperforms PQDiff with smaller ones. For 5x and 11.7x experiments, the size of extrapolated images is much larger than the given input sub-image. As a result, PQDiff with smaller crop ratios outperforms PQDiff with larger ones. On top of that, we also find when the anchor crop ratio equals 0.50, the inception score drops with a large range, and we guess that is because when the random crop ratio is set $(0.50, 0.50)$, we always feed the images with the same scales to PQDiff, and correspondingly, PQDiff can not learn to scale images. Then, in the sampling stage, since images in the test set are usually in different scales, it is difficult for PQDiff to handle the scaling gaps between the training stage and the testing stage.

Table 5: Performance of plain diffusion models and PQGAN (integrate PQ scheme in GAN) on the Scenery dataset without the "copy" operation (as we report the center PSNR score).

| Method | 2.25x | | | 5x | | | 11.7x | | |
|---|---|---|---|---|---|---|---|---|---|
| | FID ↓ | IS ↑ | Center PSNR ↑ | FID ↓ | IS ↑ | Center PSNR ↑ | FID ↓ | IS ↑ | Center PSNR ↑ |
| Valinna diffusion (Ho et al., 2020) | 57.425 | 3.816 | 11.538 | 59.475 | 3.286 | 10.894 | 73.587 | 2.867 | 10.347 |
| PQGAN (Ours) | 32.139 | 3.811 | 27.097 | 38.466 | 3.216 | 26.893 | 49.943 | 2.853 | 24.139 |
| PQDiff (Ours) | 29.446 | 3.849 | 27.676 | 34.492 | 3.547 | 27.267 | 44.517 | 3.269 | 24.797 |

Table 6: Impact of different types of positional embeddings on Scenery dataset. All methods are without "copy" operation. The sin-cos positional embedding is also used in the main experiments.

| Method | 2.25x | | | 5x | | | 11.7x | | |
|---|---|---|---|---|---|---|---|---|---|
| | FID ↓ | IS ↑ | Center PSNR ↑ | FID ↓ | IS ↑ | Center PSNR ↑ | FID ↓ | IS ↑ | Center PSNR ↑ |
| None | 61.382 | 3.811 | 10.382 | 75.281 | 3.298 | 10.285 | 90.271 | 2.974 | 10.286 |
| Learnable | 32.141 | 3.816 | 27.263 | 37.971 | 3.387 | 26.885 | 49.136 | 3.179 | 23.461 |
| Sin-Cos (Eq.1) | 29.446 | 3.849 | 27.676 | 34.492 | 3.547 | 27.267 | 44.517 | 3.269 | 24.797 |

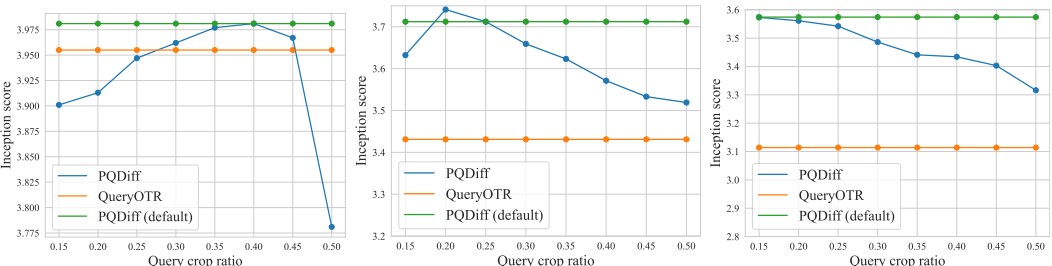

Figure 5: Inception scores on the Scenery dataset with different random crop ratios of anchor view $\mathbf{x}_a$ over 2.25x (left), 5x (middle), and 11.7x (right). In some cases, PQDiff outperforms PQDiff (Default), as we do not heavily tune the hyper-parameters in the default version. The query crop ratio is randomly sampled from $r \sim 0.5$, where $r$ is the values in the horizontal axis of the plots.

**Integrate PQ scheme into other generating models.** Beyond diffusion, we also integrate our PQ learning paradigm into GAN-based model. In addition, to analyze the effect of the positional query, we also report the performance when directly using diffusion models (Ho et al., 2020). Table 5 shows the results. Specifically, we find the quality of the images generated by GAN is lower than the diffusion model (Ho et al., 2020), as the inception score of PQGAN is lower than diffusion models (Ho et al., 2020). However, the diffusion model is not well conditioned by the input sub-image (the FID and Center PSNR scores are much worse than PQGAN and PQDiff). The high FID and Center PSNR scores indicate the proposed PQ scheme can provide a strong condition, enhancing the generative models (Goodfellow et al., 2014; Ho et al., 2020) to learn where to outpaint.

**Impact of the Positional Embedding.** To analyze the effect of the positional embedding, we conduct a group of experiments with different types of positional embeddings. We mainly consider two types of embeddings (sin-cos and learnable). Learnable embedding means we take the relative position $(m, n)$ as input and use an MLP composed of two linear layers with the activation function to map the 2-dimensions to $D$-dimensions. Table 6 shows the results with different positional embedding. Note that None means without relative positional embedding. Thus, the model can not learn the positional relationships between input sub-image and extrapolated images. Correspondingly, the inception score only drops with a little range, but the FID and Center PSNR drop with a large range.

## 5  CONCLUSION

We have proposed PQDiff, which learns the positional relationships and pixel information at the same time. Methodically, PQDiff can outpaint at any multiple in only one step, greatly increasing the applicability of image outpainting. We conduct experiments on three standard outpainting datasets, where PQDiff achieves new SOTA results that surpass previous methods by a large margin under almost all settings. We also conduct comprehensive ablation studies to show the robustness of our approach, including crop ratios, the center PSNR score, and the relative positional embeddings.

**Ethics Statement.** PQDiff generates images conditioned by positional embeddings, learning pixel information and positional relationships simultaneously. As the datasets used in PQDiff primarily

focus on scenery, buildings, and arts, there are currently minimal negative potential impacts on ethics and crime-related aspects. We are aware that any technology could be abused for ill purposes.

**Reproducibility Statement.** We have clarified training and sampling details including hyper-parameters, pseudo code of the relative positional embeddings, and training pipeline in Sec. B in the Appendix. In addition, all the datasets used in this paper are open-source and can be accessed online.

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

# Supplementary Material

## A    PSEUDO CODE OF THE RELATIVE POSITIONAL EMBEDDINGS

Pytorch-liked code of the relative positional embeddings.

```python
def get_views_and_rpe(pil_image, scales, ratios, size):
    """
    :param pil_image: images with PIL format
    :param scales pil_image: random crop scale
    :param ratios pil_image: random crop ratios
    :param size: size of cropped views
    """
    # Calculate the position of the anchor view
    i_a, j_a, h_a, w_a = RandomResizedCrop.get_params(pil_image, scales,
        ratios)
    anchor_view = resized_crop(pil_image, i_a, j_a, h_a, w_a, (size,
        size))
    # Calculate the position of the target view
    i_t, j_t, h_t, w_t = RandomResizedCrop.get_params(pil_image, scales,
        ratios)
    target_view = resized_crop(pil_image, i_t, j_t, h_t, w_t, (size,
        size))
    # Derive the relative positions of two views
    grid = calculate_sin_cos([i_a, j_a, h_a, w_a], [i_t, j_t, h_t, w_t])
    # embed_dim is relevant to the model, which is predefined
    pos_embed = get_2d_sincos_pos_embed_from_grid(embed_dim, grid)
    return pos_embed

def calculate_sin_cos(anchor_pos, target_pos, anchor_grid_size=14,
    target_grid_size=14):
    """
    :param target_pos: [i_t, j_t, h_t, w_t]
    :param anchor_pos: [i_a, j_a, h_a, w_a]
    :param target_grid_size: sequence length of the target view (relative
        to patch size of models)
    :param scales pil_image: sequence length of the anchor view (relative
        to patch size of models)
    :return grid: 2-d grid of the relative positions
    """
    kg = anchor_pos[3] / anchor_grid_size
    # calculate bias of width
    w_bias = (target_pos[1] - anchor_pos[1]) / kg
    kl = target_pos[3] / target_grid_size
    # calculate scales of width
    w_scale = kl / kg
    kg = anchor_pos[2] / anchor_grid_size
    # calculate bias of height
    h_bias = (target_pos[0] - anchor_pos[0]) / kg
    # calculate scales of height
    kl = target_pos[2] / target_grid_size
    h_scale = kl / kg
    grid_h = np.arange(h_bias, grid_size * h_scale + h_bias-5e-3,
        h_scale, dtype=np.float32)
    grid_w = np.arange(w_bias, grid_size * w_scale + w_bias-5e-3,
        w_scale, dtype=np.float32)
    # make the width and height grids, and width goes first
    grid = np.meshgrid(grid_w, grid_h)
    grid = np.stack(grid, axis=0)
    grid = grid.reshape([2, 1, grid_size, grid_size])
    return grid
```

# B  TRAINING DETAILS

**Datasets.** We use three datasets without any data selection strategies Tan et al. (2024): Scenery (Yang et al., 2019), Building Facades (Gao et al., 2023), and WikiArt (Tan et al., 2016), in line with (Yang et al., 2019; Yao et al., 2022; Van Hoorick, 2019). **Scenery** is a natural scenery with diverse natural scenes, consisting of about 5,000 images for training and 1,000 images for testing. **Building Facades** is a city scenes dataset consisting of about 16,000 and 1,500 images for training and testing, respectively. **WikiArt** is a fine-art paintings dataset, which can be obtained from wikiart.org. We use the split manner of genres datasets (used in (Yao et al., 2022; Gao et al., 2023)), which contain 45,503 training images and 19,492 testing images.

**Training Details.** We implement our approach with PyTorch (Paszke et al., 2019) on a platform equipped with 8 V100 GPUs. The encoder is composed of 8-10 stacked transformer blocks. Then, a cross-attention block is employed, followed by the decoder made of 8-10 transformer blocks. Finally, a 3x3 convolutional layer is adopted to smooth the generated image. In line with previous methods (Yao et al., 2022), we copy the ground truth (input) to the corresponding location in the generated image. We find the copy operation will make a great impact on previous methods, while PQDiff is much more robust to this operation, which we will discuss later. The number of parameters is approximately equal to QueryOTR (Yao et al., 2022), which contains 12 transformer blocks in the encoder and 4 transformer blocks in the decoder. We adopt AdamW optimizer (Loshchilov & Hutter, 2019), and we set the learning rate to 0.0002, weight decay to 0.03, and betas to 0.99. In the training stage, we set the random crop ratio of the anchor view to (0.15, 0.5) and the ratio of the target view to (0.8, 1.0), aiming to use the small view to predict the larger view. We train PQDiff 80k, 150k, and 300k iterations with 64 images per GPU on the Scenery, Building Facades, and WikiArt datasets, respectively. Following QueryOTR, we set the resolution of each cropped view as 192x192. Our core idea of PQ-Diff is to utilize the randomly cropped two views (anchor and target) to learn the positional relation between them.

**Evaluation and Baselines.** We use Inception Score (IS) (Salimans et al., 2016), Frechet Inception Distance (FID) (Heusel et al., 2017) to measure the generative quality. Note that the upper bounds of IS are 4.091, 5.660, and 8.779 for Scenery, Building Facades, and WikiArt, respectively, which are calculated by real images in the test set. Here we do **not** use PSNR (peak signal-to-noise ratio) as once used in outpainting because PSNR cannot reflect the **diversity** of generated images, which is important for generative models, and more details are given in Sec. 4.2. Alternatively, we report the PSNR score between the input sub-images and the center region of the generated images. Because we think this score is more meaningful to detect whether the network ignores the input conditions. We make comparisons with five SOTA image outpainting methods, NSIPO (Yang et al., 2019), SRN (Wang et al., 2019), IOH (Van Hoorick, 2019), Uformer (Gao et al., 2023) and QueryOTR (Yao et al., 2022).

**Sampling Details.** In line with the previous method (Yao et al., 2022; Gao et al., 2023), for the testing stage, all images are resized to 192x192 as the ground truth, and then the input images are obtained by center cropping to the sizes 128x128, 86x86, and 56x56 for 2.25x, 5x, and 11.7x outpainting, respectively. The total output sizes are 2.25, 5, and 11.7 times the input in terms of 2.25x, 5x, and 11.7x outpainting, respectively.

**Algorithm 1:** Training pipeline of PQDiff.

**Input:** Raw complete image $\mathbf{x}$, the generating network for training: $g_\theta$, VQVAE encoder and decoder $F_E, F_D$, timestep $T$.

**Result:** The pretrained network $g_\theta$.

1 Initialize $g_\theta$.
2 **repeat**
3     Randomly sample $t$ from $1 \sim T$.
4     $\mathbf{x}_a, \mathbf{x}_b \leftarrow$ RandomCropResize($\mathbf{x}$).
5     Compute the relative position $\mathbf{E}$.
6     Obtain latent embeddings $\mathbf{z}_{a,b} \leftarrow F_E(\mathbf{x}_{a,b})$.
7     Randomly sample Gaussian noise $\epsilon_t$.
8     Add noise $\epsilon_t$ to obtain $\mathbf{z}_{b_t}$ via Eq.2.
9     Predict $\widetilde{\epsilon}_t$ by $g_\theta(\mathbf{z}_a, \mathbf{z}_{b_t}, t)$.
10     Compute the loss $\|\widetilde{\epsilon}_t - \epsilon_t\|_p^p$ via Eq.3.
11     Backward and update the network $g_\theta$.
12 **until** *Converge*;

**Output:** $g_\theta$

**Algorithm 2:** Sampling pipeline of PQDiff.

**Input:** Input sub-image $\mathbf{x}$, the pretrained network $g_\theta$, VQVAE encoder and decoder $F_E, F_D$, timestep $T$, the specified outpainting multiple $K$.

**Result:** Outpainted image $\mathbf{y}$.

1 Compute the relative position $\mathbf{p}$ of $\widetilde{\mathbf{y}}$ through $K$ (illustrated in Fig. 2).
2 Acquire the positional embedding $\mathbf{E}$ through $\mathbf{p}$.
3 Randomly sample Gaussian noise $\mathbf{y}_T$.
4 **repeat**
5     Acquire latent embeddings $\mathbf{z} \leftarrow F_E(\mathbf{x})$.
6     Predict $\widetilde{\epsilon}_T$ by $g_\theta(\mathbf{z}_a, \mathbf{y}_T, T)$.
7     Compute $\widetilde{\mathbf{y}}_0$ and $\mathbf{y}_{T-1}$ through Eq. 5.
8     $T \leftarrow T - 1$.
9 **until** $T \geq 1$;
10 $\mathbf{y} \leftarrow F_D(\mathbf{y}_0)$.

**Output:** $\mathbf{y}$.

## C ILLUSTRATION OF THE RPE DURING THE TRAINING AND SAMPLING

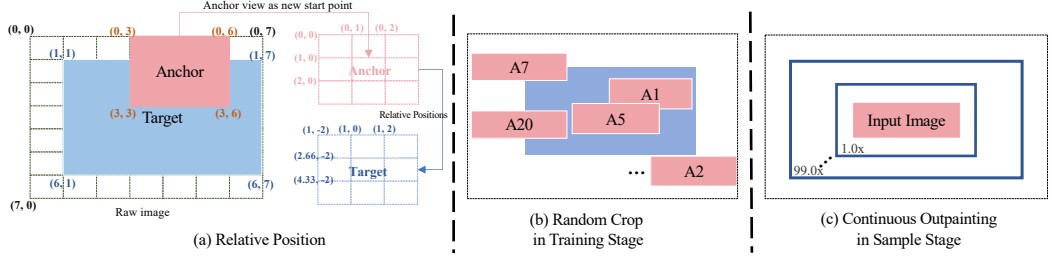

Figure 6: Here is the detailed working mechanism of Relative Position Embeddings (RPE). Among them are (a) Specific calculation example diagram of RPE. (b) RPE captures different position relationships by randomly sampling the anchor view and target view during the training stage. (c) RPE achieves continuous outpainting through the learned positional relation in the sample stage.

**Relative positional embeddings.** Instead of simply using learnable positional embeddings (commonly used in previous transformer-based learning methods (Gao et al., 2023)), which can not represent the relation between the anchor view and target view (as the positional relation is random for each sample at each iteration due to the randomness of the *RandomCrop* augmentation), we adopt fixed positional encoding to represent the relative positions between the anchor view and each query tokens (i.e., each relative positional token in the target view), which is illustrated in Fig. 6. Given the positions $\mathbf{p}_A = \{p_{Ai}, p_{Aj}, p_{Ah}, p_{Aw}\}$ (top location, left location, height, and width), $\mathbf{p}_T = \{p_{Ti}, p_{Tj}, p_{Th}, p_{Tw}\}$ of the two views $\mathbf{x}_A$ and $\mathbf{x}_T$, the objective is calculating the relative position of each patch $[\mathbf{x}_T^{(1)}, \mathbf{x}_T^{(2)}, \cdots, \mathbf{x}_T^{(K_T^2)}]$ in the target view based on the anchor view $\mathbf{x}_A$, where $(K_T^2)$ is the number of patches in the target view. Specifically, we calculate the patch-level relative position of the target view via the following equation:

$$p_t^{m,n} = [p_t^m, p_t^n] = [\underbrace{\frac{K_a \cdot (p_{ti} - p_{ai})}{p_{ah}} + \frac{p_{th} \cdot K_a \cdot (m-1)}{K_t \cdot p_{ah}}}_{Row}, \underbrace{\frac{K_a \cdot (p_{tj} - p_{aj})}{p_{aw}} + \frac{p_{tw} \cdot K_a \cdot (n-1)}{K_t \cdot p_{aw}}}_{Column}]$$

(8)

where $K_a^2$ means the number of patches of the anchor view. $p_t^{m,n}$ means the relative position of the patch located at $m$-th row and $n$-th column in the target view. Then, for each patch, we use the

following popular form in transformers (Vaswani et al., 2017) to generate the relative positional embedding:

$$\mathbf{P}_T^{m,n} = \left[ \sin\left(\frac{p_t^m}{e^{2*1/d}}\right), \cos\left(\frac{p_t^m}{e^{2*2/d}}\right), \cdots, \sin\left(\frac{p_t^m}{e}\right), \right.$$
$$\left. \sin\left(\frac{p_t^n}{e^{2*1/d}}\right), \cos\left(\frac{p_t^n}{e^{2*2/d}}\right), \cdots, \sin\left(\frac{p_t^n}{e}\right) \right] \tag{9}$$

where $\mathbf{P}_T^{m,n} \in \mathbb{R}^{1 \times D}$ is the relative positional embeddings of the patch located at $m$-th row and $n$-th column in target view. $D$ is the hidden dimension of the model and $D = 2 * d$. $e$ is the pre-defined parameter that will be universally set to 10,000 in our experiments, in line with MAE (He et al., 2022). $(m, n)$ means the position of the top left patch position (as described in Eq. 8 and illustrated in Fig. 6).

## D    MORE RESULTS IN CONTINUOUS MULTIPLES

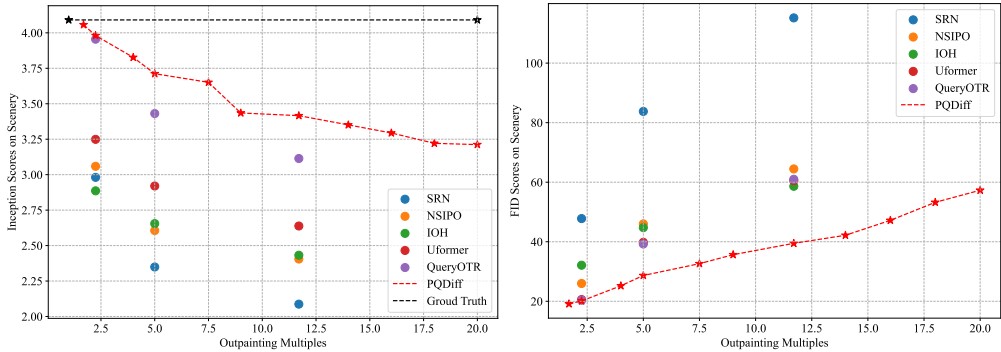

Figure 7: Comparisons between PQDiff and previous methods v.s. different outpainting multiples, where our PQDiff achieves state-of-the-art results under almost all settings. Note that since PQDiff can additionally outpaint images with continuous multiples, we show the results of PQDiff as a continuous curve instead of scatter plots.

## E    IMPACT OF THE CONTINUOUS AND DISCRETE POSITIONAL EMBEDDINGS

To further explore how PQDiff learns the relative position, we conduct a set of ablation studies. Specifically, we remove the randomness of the positions of the anchor views and target views. Specifically, the anchor view is the center region of the target view, and the target view is set 2.25x, 5x, and 11.7x larger than the anchor view with the same probability at each training iteration (1/3 for each multiple) to better compare with the main results in our paper. Therefore, the sizes of the anchor view and target view are discrete, as they are fixed as three numbers. We train the Discrete version on 8 V100 GPUs with 80k iterations (in line with the main results). Table 7 shows the results when evaluating with variant outpainting multiples. We find under 2.25x, 5x, and 11.7x settings (inner distribution), discrete training, and achieve similar results with PQDiff. However, when transferred to other outpainting multiples (4x, 9x, 16x), both three scores of discrete training drop, especially for Center PSNR. We guess the reason behind the interesting phenomenon is that the "Discrete training" strategy only memorizes the pixel information of the input sub-image, but fails to scale the sub-image with proper ratio due to the outpainting multiples gaps in the training and inference stage.

| Metric | Method | Inner dist | | | Outer dist | | |
|---|---|---|---|---|---|---|---|
| | | 2.25x | 5x | 11.7x | 4x | 9x | 16x |
| IS | Discrete | 3.813 | **3.553** | 3.207 | 3.626 | 3.391 | 2.891 |
| | PQDiff (Ours) | **3.849** | 3.547 | **3.269** | **3.827** | **3.435** | **3.012** |
| FID | Discrete | 29.481 | **33.917** | 47.239 | 35.183 | 50.183 | 53.174 |
| | PQDiff (Ours) | **29.446** | 34.492 | **44.517** | **31.274** | **39.927** | **49.271** |
| Center PSNR | Discrete | **27.814** | 27.017 | 23.917 | 18.271 | 17.238 | 14.927 |
| | PQDiff (Ours) | 27.676 | **27.267** | **24.697** | **27.477** | **25.820** | **23.175** |

Table 7: Performance when transferring the model to different evaluation distributions. "Inner dist" means evaluating the model under the same settings as the training stage. "Outer dist" means evaluating the model under different settings with the training stage.

## F  SAMPLING SPEED

For better comparisons, we provide results of PQDiff with more timesteps in Table 8. Specifically, in the training phase, we set timesteps as 1,000. In the testing phase, we change the timesteps from 50∼500 (since we observe with larger timesteps, the IS and FID score won't improve anymore). We find when timesteps are set to 300, PQDiff achieves 4.203 IS scores, while ground truth only achieves 4.091, which further demonstrates the vividness of images generated by PQDiff.

Table 8: Comparison of sampling time, FID, and inception scores with 2.25x, 5x, 11.7x settings on Scenery dataset. The sampling time is the wall-clock time spent on generating 64 images on 8 V100 GPUs. The best results and second best results are **boldface** and underlined, respectively.

| Method | 2.25x | | | 5x | | | 11.7x | | |
|---|---|---|---|---|---|---|---|---|---|
| | Time (Sec.) | FID ↓ | IS ↑ | Time (Sec.) | FID ↓ | IS ↑ | Time (Sec.) | FID ↓ | IS ↑ |
| PQDiff (50 timesteps) | **12.175** | 20.103 | 4.046 | **12.175** | **27.051** | 3.710 | **12.175** | 41.242 | 3.741 |
| PQDiff (100 timesteps) | 24.481 | 20.878 | 3.968 | 24.481 | 28.424 | 3.719 | 24.481 | **39.194** | 3.496 |
| PQDiff (200 timesteps) | 49.021 | 19.838 | 4.111 | 49.021 | 31.039 | 3.940 | 49.021 | 40.265 | 3.583 |
| PQDiff (300 timesteps) | 73.586 | 21.072 | **4.203** | 73.586 | 28.856 | 3.957 | 73.586 | 39.607 | 3.479 |
| PQDiff (400 timesteps) | 98.101 | 20.623 | 4.028 | 98.101 | 28.939 | 3.761 | 98.101 | 39.618 | 3.508 |
| PQDiff (500 timesteps) | 122.739 | **19.828** | 4.098 | 122.739 | 27.561 | 3.728 | 122.739 | 41.066 | 3.709 |
| Ground Truth | ∼ | ∼ | 4.091 | ∼ | ∼ | **4.091** | ∼ | ∼ | **4.091** |

## G  IMPACT OF PREDICTING $x_0$ OR NOISE.

PQDiff can also be thought of as a predictive task conditioned by the relative information and the anchor views. Hence, we conduct the experiments to directly predict $\mathbf{x}_b$, which is similar to QueryOTR (Yao et al., 2022), and we report the inception score in Table 9. We find by predicting $\mathbf{x}_b$, PQDiff is much worse than predicting noise under all settings (especially in the 11.7x setting), and the phenomenon is also consistent with previous generative tasks (Bao et al., 2023; Ho et al., 2020). We guess that's because predicting $\mathbf{x}_b$ makes the task more predictive but not generative, and the learned network will overfit in the training set, resulting in bad generalization for the generative task in the test set.

## H  INCORPORATE WITH PRETRAINED MODELS

As shown in Tab. 10 on the scenery dataset, We consider adding an ablation study to analyze the usage of the pretrained model. First, we try to load the stable diffusion pretrained model in PQDiff. As we use VQVAE and cross attention in our model (our model is a transformer-based model, while stable diffusion mainly uses resnet-block), we can only load weights of the spatial transformer layer.

Then, we try to train our model on ImageNet with 80,000 iterations, and then, we re-train the model on the Scenery dataset. We can find that pretrained with the imagenet has improved consistency, which provides insights for the subsequent scale-up of our framework. We can find that pretrained with the Imagenet has improved consistency, which provides insights for the subsequent scale-up of our framework.

## I    MORE GENERATED EXAMPLES ON FACADE DATASETS

We put more generated examples on facade datasets in Fig. 9. We have observed two interesting phenomena when our PQDiff extends semantic structures in facade scenes.

- Our PQDiff can expand the reliable semantic structure that aligns with human cognition. In contrast, the previous SOTA QueryOTR model pretends to generate a physically distorted and abnormally deformed semantic structure. (As shown in the middle column (Fig. 9), the comparison includes the First row(lane), second row(wooden building), and third row (teaching buildings).

- Our PQDiff can expand more texture semantic details in facade scenes, thereby generating high-fidelity expanded images. However, QueryOTR will lose detailed information, and the generated extended image has a lot of noise. (Representative examples are shown in the first column (First row, second row, and third row).

## J    ABSOLUTE POSITION EMBEDDING V.S. RELATIVE POSITION EMBEDDING

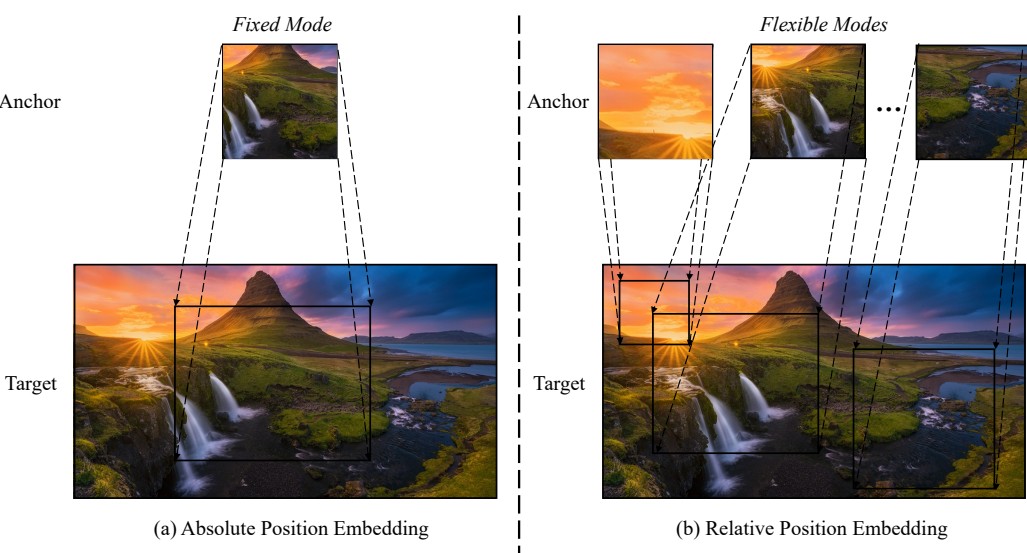

Figure 8: The difference: (a) Absolute Position Embeddings and (b) Relative Position Embeddings.

Table 9: Comparison of $x_0$ prediction and noise prediction on Scenery dataset. Since the copy operation would lead to the infinite value in the center PSNR score, we only report the center PSNR score without the copy operation. 2.25x, 5x, and 11.7x follow the setting in the main experiments.

| Method | 2.25x | | | 5x | | | 11.7x | | |
|---|---|---|---|---|---|---|---|---|---|
| | FID ↓ | IS ↑ | Center PSNR ↑ | FID ↓ | IS ↑ | Center PSNR ↑ | FID ↓ | IS ↑ | Center PSNR ↑ |
| QueryOTR (Yao et al., 2022) | 20.366 | 3.955 | ∼ | 39.237 | 3.431 | ∼ | 60.977 | 3.114 | ∼ |
| $x_0$ pred | 37.645 | 3.353 | 26.478 | 52.505 | 3.230 | 26.027 | 69.406 | 2.984 | 23.835 |
| $x_0$ pred + Copy | 27.356 | 3.920 | ∼ | 43.877 | 3.676 | ∼ | 67.170 | 3.556 | ∼ |
| Noise pred | 29.446 | 3.849 | **27.676** | 34.492 | 3.547 | **27.267** | 44.517 | 3.269 | **24.797** |
| Noise pred + Copy | **20.100** | **3.981** | ∼ | **28.668** | **3.712** | ∼ | **39.465** | **3.574** | ∼ |

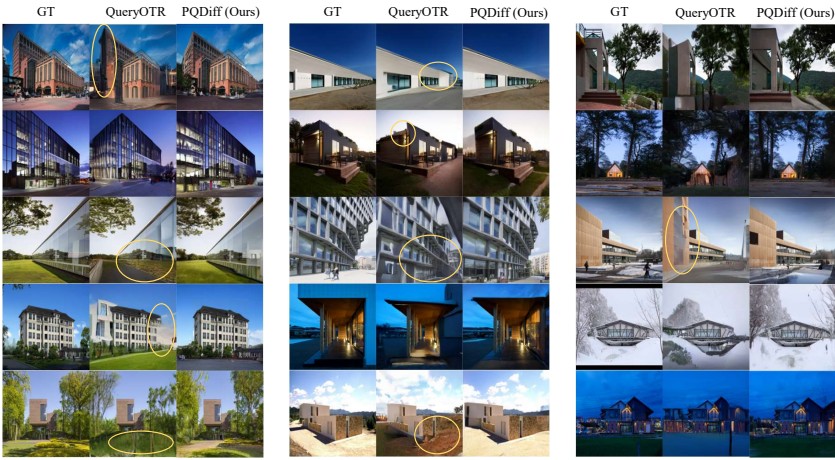

Figure 9: Comparisons of PQDiff with the SOTA method QueryOTR on the facades building dataset. We highlight the noise and spots generated by QueryOTR with **yellow** boxes.

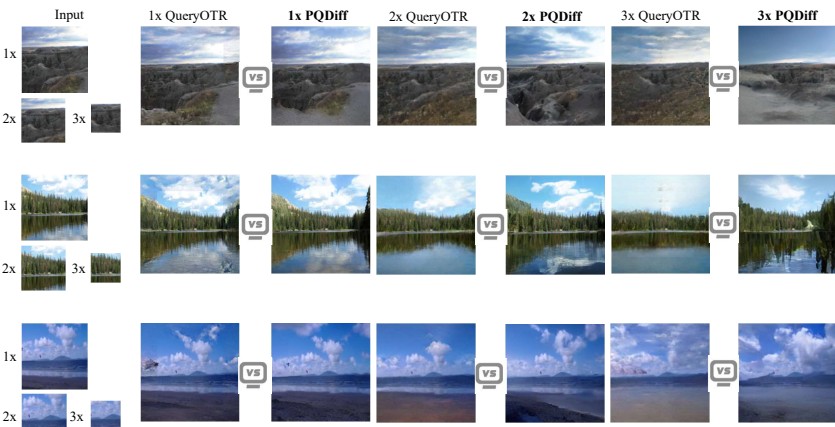

Figure 10: Qualitative comparisons with the previous SOTA method QueryOTR (Yao et al., 2022).

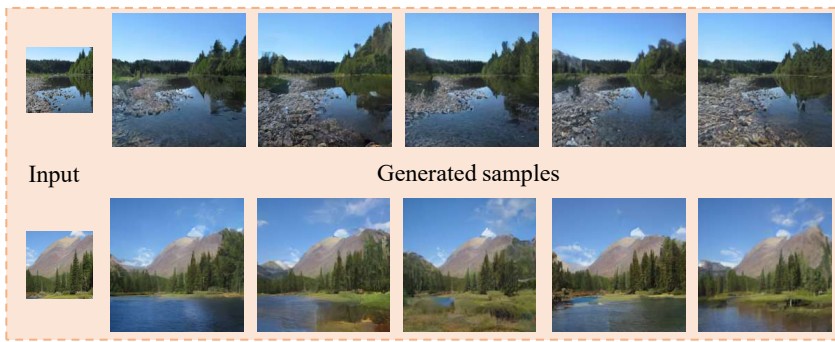

Figure 11: Example of images generated by PQDiff with fixed positions.

Table 10: The experiment that incorporates our PQDiff with pretrained stable diffusion.

| Method | 2.25x | | 5x | | 11.7x | |
|---|---|---|---|---|---|---|
| | FID ↓ | IS ↑ | FID ↓ | IS ↑ | FID ↓ | IS ↑ |
| w/o pretrain | 20.100 | 3.981 | 28.668 | 3.712 | 39.465 | 3.574 |
| w SD pretrain | 20.164 | 3.985 | 28.537 | 3.692 | 38.917 | 3.619 |
| w Imagenet pretrain | **19.791** | **3.993** | **28.016** | **3.731** | **36.271** | **3.629** |

# K   MORE CASES GENERATED BY PQDIFF

We show more examples generated by PQDiff in Fig. 12, Fig. 13 and Fig. 14.

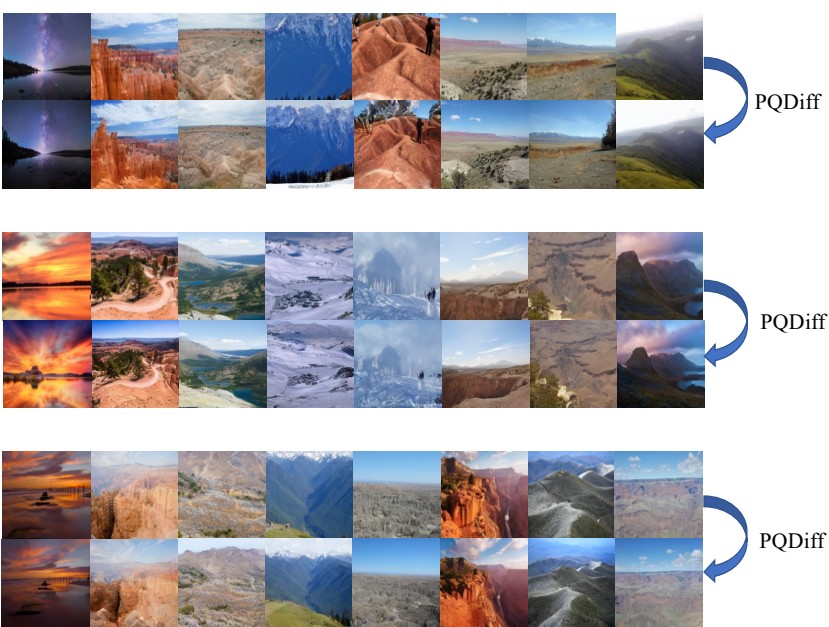

Figure 12: Example of images generated by PQDiff in the testing set.

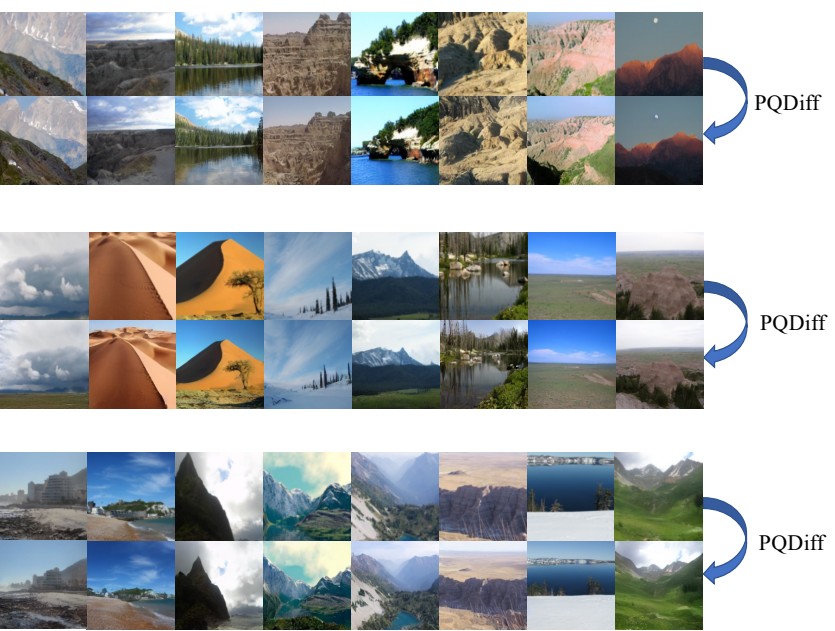

Figure 13: Example of images generated by PQDiff in the testing set.

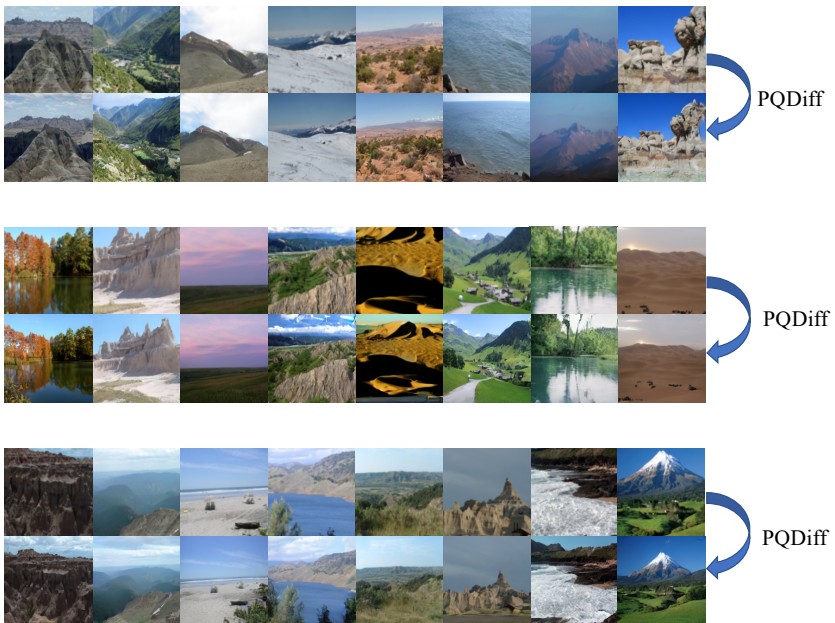

Figure 14: Example of images generated by PQDiff in the testing set.

## L    MORE DISCUSSIONS

**Broader impact and potential applications.**    As PQDiff can generate images conditioned by arbitrary relative positions, we believe our method has great potential in image-inpainting (change the RPE of the target view with the inpainting regions), and super-resolutions (interpolate the relative position of the anchor view) if we properly change the relative positional in the training and sampling stages.

