# OpenReview forum: "Continuous-Multiple Image Outpainting in One-Step via Positional Query and A Diffusion-based Approach"
_ICLR.cc/2024/Conference — ICLR 2024 poster_

### Official Review · Reviewer_xgEg · 2023-10-31

**Soundness:** 3 good
**Presentation:** 2 fair
**Contribution:** 3 good
**Rating:** 6
**Confidence:** 5

**Summary:**

This paper proposes an diffusion-based image outpainting model that can outpaint images with arbitrary and continuous multiples. This is achieved with the help of the relative positional embedding that can encode any relative position relations. Experimental results validate the effectiveness of the proposed model.

**Strengths:**

1. The target of outpainting images with arbitrary and continuous multiples is very attractive and highly practical.
2. Using relative positional embedding to deal with the problem of arbitrary and continuous multiples is simple but effective.

**Weaknesses:**

1. The main weakness of this paper is some unclear explanations and descriptions.

(1) The authors categorize existing methods into two classes, GAN-based and MAE-based methods, and emphasize that one of their  limitations is  the multiple running times to achieve the large expansion multiple. However, it is unclear why these two types of methods need to run multiple times. Using GAN or MAE is not the crux.

(2) As the authors fail to clearly explain why previous methods cannot outpaint images with arbitrary and continuous multiples. The strengths and the contributions of the proposed method become unclear and unconvincing.

(3) The authors say "to achieve one-step diffusion-based generation, we propose to use relative positional queries and input sub-images as conditions". This is really strange. Why relative positional queries can reduce the number of diffusion steps?

(4) In section 3, the authors say that they set h1 = h2 = h and w1 = w2 = w for notation simplicity. It looks improper.

(5) Some parts of Figure 2 are confusing. For example, in the training part, the relative positional embedding takes the two image views as inputs. But Eq.1 shows that only the relative position (m,n) is needed to generate the embedding. The images are not the inputs of it. In the sampling part, it is unclear what the inputs are. I cannot know which of the original image, the cropped image, and the masked image are the inputs

(6) In Eq.2, it seems Xb0 should be Zb0

2. There is a strange point in the proposed model. As shown in Figure 2, the relative position only reflects the position relations of the top-left corners between the two views. In other words, even the anchor view and the relative position are fixed, there can still be many different target views. How does this one-to-many mapping influence the model? Furthermore, in the sampling stage, if the input noise, image and relative position are all fixed, how to ensure the outpainting ratios on four edges are the same as what we expected?

**Questions:**

The questions and confused points are given in Weakness Section. I am looking forward to the responses from the authors. I may improve my rating if the questions have satisfactory answers.

---

> ### Author Response · Authors · 2023-11-13
> **Response to Reviewer xgEg (Part 1)**
>
> Thank you for the time, thorough comments, and nice suggestions. We are pleased that you acknowledged the novel idea, technical soundness, and effective experiments.
>
> > **Q1**:  Some unclear explanations and descriptions.
>
> **A1**: Thanks for your detailed review, here are our clarifications.
>
> > **Q1.a**: It is unclear why these two types of methods (GAN or MAE) need to run multiple times.
>
> **A1.a**: (1) For previous methods, in their training stage, they outpaint images from the given central image 128x128 to 192x192, and the training settings are fixed. As a result, there are 2.25 ($\frac{128}{192}^2$) times more contents are generated.
>
> (2) However, If we want to generate 57.6 ($\approx(2.25)^{5}$) times more contents, we have to forward the pretrained generative network 5 times ($1$->$f(x,)$->$2.25$-$f(x,)$->$(2.25)^{2}$->resize->$f(x,)$->$(2.25)^{3}$->resize)->$f(x,)$->$(2.25)^{4}$->resize->$f(x,)$->$(2.25)^{5}$).
>
> (3) The reason for categorizing the existing methods into GAN-based and MAE-based is because GAN-based methods are difficult to converge and MAE-based methods are relatively more efficient due to the dropping of masked patches.
>
> > **Q1.b**: The authors fail to clearly explain why previous methods cannot outpaint images with arbitrary and continuous multiples.
>
> **A1.b**: **Arbitrary and continuous:** For previous methods, they train their model by inputting the central sub-image, and the outpainting multiples (e.g., 1.5x) are pre-fixed in the training stage. Then, in the sampling stage, if we want to outpaint the image 1.7x, they have to retrain their model again. However, our PQDiff is much more flexible. Once PQDiff finishes training, PQDiff can generate images with any multiples.
>
> > **Q1.c**: Why relative positional queries can reduce the number of diffusion steps?
>
> **A1.c**: The relative positional queries do not reduce the number of diffusions **timesteps**, but reduce the number of **iterations** for large multiple settings. As described in A1.a, previous methods have to forward the network 5 times to obtain 57.6x times more content. In our methods, for any outpainting multiples $n$, we only forward the network one times $1$->$f(x)$->$(2.25)^{n}$. Note that $f(x)$ includes all the diffusion steps.
>
> > **Q1.d**: In section 3, the authors say that they set h1 = h2 = h and w1 = w2 = w for notation simplicity. It looks improper.
>
> **A1.d**: Thanks for your nice suggestions. Actually, the regions of the target view are larger than the regions of the anchor view. $h1 = h2 = h$ and $w1 = w2 = w$ are because we use the resize operation to conveniently feed the two views into transformers. We have corrected this expression in our new version.
>
> > **Q1.e**: The mismatch of the training and sampling in Figure 2 and Eq.1.
> >
> **A1.e**: There are some misunderstandings in the training stage. Actually, in the training stage, we feed the $z_{a}$, noised $z_{b_{t}}$ and relative positional embeddings as input. Please note $z_{b_{t}}$ is relatively equivalent to $x_{t}$ in the original diffusion model, which can be calculated by diffusion forward process (see line 9 in Alg.1 in Appendix B).
> Then, **for the sampling stage**, the inputs are the given sub-image (anchor image) $z_{a}$ , a randomly initialized Gaussian noise, and relative positional embeddings.
> For better understanding, we have modified the Fig.2. Please check our new version.
>
> > **Q1.f**: In Eq.2, it seems Xb0 should be Zb0 In Eq.2, it seems Xb0 should be Zb0
>
> **A1.f**: Thanks for your detailed review. We have corrected the typo in our new version.
>
> > **Q2**:  How does this one-to-many mapping influence the model?
>
> **A2**: We are sorry for the confusion of the figure, and actually, the anchor-to-target is a one-to-one mapping. Since the relative position is one of the **core parts** of our method, we hope our clarification could help your understanding:
>
> Instead of using image-level relative position, we will record the top-left positions for **each patch** of the target view, which means that **the target view is determined**. Take an example:
>
> in the training stage, we take the anchor view as the starting point, suppose the positions of the anchor patches are:
> $$\begin{pmatrix}
> (0,0)&(0,1)&(0,2)\\\\
> (1,0)&(1,1)&(1,2)\\\\
> (2,0)&(2,1)&(2,2)\\\\
> \end{pmatrix},$$ and the relative positions of the target patches are:
> $$\begin{pmatrix}
> (-1.6,1.7)&(-1.6,3.4)&(-1.6,5.1)\\\\
> (1.4,1.7)&(1.4,3.4)&(1.4,5.1)\\\\
> (4.4,1.7)&(4.4,3.4)&(4.4,5.1)\\\\
> \end{pmatrix}.$$ From the two positions, we know the top-left **shift bias** of the anchor and target view is $[-1.6-0, 1.7-0]$, the **shift scaling** of the anchor and target view is $[(1.4-(-1.6))/(1-0), (3.4-1.7)/(1-0)]$. Then, the position of the target view is certainly determined.
> We also illustrate the details of how to calculate relative positional (both formulation and illustration) in **Appendix C** in our new version.

---

> ### Author Response · Authors · 2023-11-13
> **Response to Reviewer xgEg (Part 2)**
>
> > **Q3**: In the sampling stage, if the input noise, image, and relative position are all fixed, how to ensure the outpainting ratios on the four edges are the same as what we expected?
>
> **A3**: Due to the randomness of the relative positions between the anchor view and target view in the **training stage**, the model can learn almost all the positional relations between two views. Then, in the **sampling stage**, for example, if the position input sub-image is:
> $$anchorPos=\begin{pmatrix}
> (0,0)&(0,w)&(0,2w)\\\\
> (h,0)&(h,w)&(h,2w)\\\\
> (2h,0)&(2h,w)&(2h,2w)\\\\
> \end{pmatrix}.$$
> we can set the relative position of the target patches as:
> $$
> targetPos=\begin{pmatrix}
> (-h,-w)&(-h,w)&(-h,3w)\\\\
> (h,-w)&(h,w)&(h,3w)\\\\
> (3h,-w)&(3h,w)&(3h,3w)\\
> \end{pmatrix}.
> $$
> Then, there are 4x more contents in the newly generated images. Specifically, the outpainting ratio (scaling) of the height is $(h-(-h))/(h-0) = 2$, and the outpainting ratio (scaling) of the width is $(w-(-w))/(w-0)) = 2$. Then, the total outpainting ratio is $2(height)*2(width)=4$. We can also find the sub-image positions $anchorPos$ are in the central area of the calculated relative positions $targetPos$ of the target patches. Correspondingly, the PQDiff can generate images on four edges at the same time.
>
> Please let us know if there are further questions.

---

> ### Comment · Reviewer_xgEg · 2023-11-14
>
> Thanks for your detailed response. My confusions have been clarified. I have raised the my rating to 6.
>
> Regarding my question 1.c, I understand that the relative positional queries are not used to reduce diffusion timesteps. But the related statement in the 4th paragraph in Introduction is really misleading. I suggest to explain these different "step" conceptions clearer in this paragraph.

---

> ### Author Response · Authors · 2023-11-14
> **Response to Reviewer xgEg**
>
> Thank you for the quick reply and nice suggestions!
>
> We have revised our new version by using **timestep** and **step** to distinguish the "diffusion step" and the "iteration" in our new revised version. Besides, we also add a footnote on page 2 in our new version for better understanding.
>
> Please let us know if you have any further questions.

---

### Official Review · Reviewer_HK74 · 2023-10-31

**Soundness:** 3 good
**Presentation:** 3 good
**Contribution:** 3 good
**Rating:** 6
**Confidence:** 4

**Summary:**

Image outpainting is practically useful yet very challenging, and current methods often require a fixed ratio of the input and output resolution beforehand. Specifically, the devised PQDiff is a single model trained for all outpainting ratios, by introducing a position query in diffusion training. The proposed method can generate output images with arbitrary and continuous multiple. Such abilities are often missing in literature.

**Strengths:**

1) The paper provided a simple yet elegant and effective approach for the challenging and important task of image outpainting. Specifically, it can naturally ensure the outpaint images with continuous multiple, and also generates the output in a single forward pass.
2) This paper shows that the position query techniques can also be applied to GAN-based models which further expand the potential impact of the paper.
3) The experimental results propsoed in Sec. 4 are convincing, in terms of both quantative results and qualitative results. Meanwhile the method is very efficient due to its single-pass inference nature against arbitary input ratio and continuous multiple.
4) Supplemental material gives detailed empirical studies to verify the effectiveness of the approach.

**Weaknesses:**

I think the authors should further discuss the potential of their method to other potential applications beyond outpainting to further expand the potential impact of this work.

**Questions:**

Are there other methods using POSITIONAL QUERY? As current existing work part does not mention them (is there were) too much.

---

> ### Author Response · Authors · 2023-11-13
> **Response to Reviewer HK74**
>
> Thank you for the time, thorough comments, and nice suggestions. We are pleased to clarify your questions step-by-step.
>
> > **Q1**:  Further discuss the potential of their method to other potential applications beyond outpainting to further expand the potential impact of this work.
>
> **A1**: Thanks for your suggestions. We have updated the potential applications in our new version (page 22). We believe our method could be applied in image-inpainting, and super-resolutions if we properly change the relative positional embeddings in the training and sampling stages.
>
> > **Q2**:  Are there other methods using positional query? As current existing work part does not mention them (if there were) too much.
>
> **A2**: There are a few works learning about how to construct useful positional embeddings in transformers [1, 2, 3]. However, the motivation behind PQDiff is different from their methods. In the image generation area, to our best knowledge, we are the first to propose relative positional queries (RPE beyond the original region) to outpaint images arbitrarily and continuously.
>
> Please let us know if there are further questions.
>
> [1] Shiv V, Quirk C. Novel positional encodings to enable tree-based transformers[J]. NeurIPS 2019.
>
> [2] Chi T C, Fan T H, Ramadge P J, et al. Kerple: Kernelized relative positional embedding for length extrapolation[J]. NeurIPS 2022.
>
> [3] Su J, Lu Y, Pan S, et al. Roformer: Enhanced transformer with rotary position embedding[J]. arXiv preprint arXiv:2104.09864, 2021.

---

> ### Comment · Area_Chair_jkGw · 2023-11-21
>
> Reviewer HK74, did the authors' rebuttal had addressed your concerns?
>
> Please reply and post your final decision as well.
>
> AC

---

### Official Review · Reviewer_urTZ · 2023-11-01

**Soundness:** 3 good
**Presentation:** 3 good
**Contribution:** 4 excellent
**Rating:** 8
**Confidence:** 4

**Summary:**

Most image outpainting method is limited by the requirement of a pre-defined output ratio before training the models. The authors propose a diffusion-based framework to solve this problem by introducing a positional query. The framework can generate output images with arbitrary and continuous ratios within one step.

**Strengths:**

- 1) The authors solve an interesting lasting problem in image outpainting research: the requirement of a pre-defined outpainting multiple.
- 2) The extensive experiments prove the superiority of the proposed framework in comparison to sota.
- 3) The paper is mostly well-written with a clear explanation of the methodology and implementation

**Weaknesses:**

1. Question regarding the wording of the contribution or the presentation in Fig. 1. See Q1.
2. More results are needed to justify the claim of continuous results. See Q2.

**Questions:**

## Questions
- 1) Is it precise to say that PQDiff handles inputs with various sizes and ratios while the output size is fixed? Assume the goal is to generate an outpainting image with a multiple of 10x given an input image of size (192, 192). Does PQDiff generate a (192, 192) version of the 10x outpainting image and have to leverage an additional decoder (perhaps super-resolution methods) to resize it to (1920, 1920)?

- 2) Can the author provide results on more multiple besides (2.5x, 5x, 11.7x) used in other sota to justify their claim of continuous multiples? For example, how is the performance of 99x? My suggestion would be a figure of FID over continuous multiples or spectrums of outpainting images with increasing multiples.

- 3) What is the quantitative evaluation of Sec 4.2 **Outpainting in an arbitrary position**, i.e. FID under different positions? Does the model prefer certain positions?

## Suggestion
- 1) The generated and input images in Supp. Sec. J (Fig. 11, 12, ...) can be put side by side for clearer comparison.

---

> ### Author Response · Authors · 2023-11-13
> **Response to Reviewer urTZ**
>
> Thanks for your constructive suggestions. Your endorsement of our method and experiments gives us significant encouragement. Here are our clarifications.
>
> > **Q1**: Is it precise to say that PQDiff handles inputs with various sizes and ratios while the output size is fixed?
>
> **A1**: Yes. As we adopt diffusion models, the output sizes are fixed. If the users want to generate images with 10x resolutions, PQDiff has to leverage an additional decoder (e.g., super-resolution).
>
> > **Q2**: Can the author provide results on more multiple besides (2.5x, 5x, 11.7x) used in other sota to justify their claim of continuous multiples. My suggestion would be a figure of FID over continuous multiples or spectrums of outpainting images with increasing multiples.
>
> **A2**: Thanks for your constructive suggestions. We have added the figure over continuous multiples with increasing multiples in our new version (see Fig.7 in the Appendix).
>
> > **Q3**: What is the quantitative evaluation of Sec 4.2 Outpainting in an arbitrary position, i.e. FID under different positions? Does the model prefer certain positions?
>
> **A3**: We conduct a set of experiments when outpainting with **arbitrary positions**. Specifically, we randomly crop the anchor view and target view, and by using RandomCropResize. The crop ratios of the anchor view and target view are $(0.4, 0.4)$ and $(0.9, 0.9)$, respectively. This setting aims to keep PQDiff generating **at least** 2.25 times contents (only when the target view contains the anchor view, the generated region are the same with centered outpainting. Otherwise, PQDiff will generate more regions than centered outpainting). Here are the results:
>
> | Method | IS | FID | Center Outpainting |
> | --- | --- | --- | ---- |
> | PQDiff + copy |  3.981 | 20.100 | $\checkmark$ |
> | PQDiff (Center position) |  3.849| 29.446 | $\checkmark$ |
> | PQDiff (arbitrary position) | 3.781 |30.129 | $\times$ |
>
> Please note that due to the random location, it's difficult to add "copy" operations for arbitrarily outpainting. We find PQDiff (arbitrary position) obtains a bit lower IS score than PQDiff (Center position), which is because the generated region of PQDiff (arbitrary position) is larger than PQDiff (Center position) on average.
>
> > **Q4**: The generated and input images in Supp. Sec. J (Fig. 11, 12, ...) can be put side by side for clearer comparison.
>
> **A4**: Thank you for your suggestion. We have put the Sec.K (Appendix) side by side for clearer comparison in our new version.
>
>
> Please let us know if there are further questions.

---

> > ### Comment · Reviewer_urTZ · 2023-11-14
> >
> > According to your response to Q1, I find Fig. 1 to be quite misleading. Since the output of different multipliers (1.0x, 3.6x, ..., 99x) should have the same resolution instead of increasing resolution. Current Fig. 1 suggests that PQDiff can generate high-res images which seems to be not true.
> >
> > Thank you for the responses to my other questions.

---

> > > ### Author Response · Authors · 2023-11-14
> > > **Response to Reviewer urTZ**
> > >
> > > Thank you for the quick reply and constructive suggestions!
> > >
> > > We have modified the Fig.1 in our new revised version. Please let us know if you have any further questions.

---

### Author Response · Authors · 2023-11-15
**General Response to ACs**

We sincerely thank the reviewers for their detailed and valuable comments. All reviewers (urTZ, HK74, xgEg) appreciate the significant experimental improvements of PQDiff and the substantial advancements it brings to the field. They highlight the proposed method is novel, simple, and elegant (HK74, xgEg). They also appreciate the excellent contribution and presentation (urTZ, HK74).

Based on these comments, we conclude some noteworthy replies for the reviewers including:

- **[Reviewer urTZ]** We have modified Fig.1 to highlight the output of different multipliers (1.0x,3.6x,...,90x) that have the same resolution in our revised version.
- **[Reviewer urTZ]** We have added the figure over continuous multiples with increasing multiples to support the claim of continuous results in our new version (see Fig.7 in the Appendix).
- **[Reviewer urTZ]** We have conducted a set of experiments and analysis when outpainting with arbitrary positions, which provides the quantitative evaluation of Sec 4.2.
- **[Reviewer HK74]** We have added the potential applications in our new version (page 10) and have discussed and cited other works that focus on the positional embeddings in transformers.
- **[Reviewer xgEg]** We have corrected this expression and the typos in our new version.
- **[Reviewer xgEg]** We have modified Fig.2 to understand how to achieve arbitrary and continuous outpainting in the sample stage.
- **[Reviewer xgEg]** We have revised our new version by using timestep and step to distinguish the "diffusion step" and the "iteration" and added a footnote on page 2 in our latest version.
- **[Reviewer xgEg]** We have clarified the anchor-to-target mapping and have illustrated the details of how to calculate relative positional (both formulation and illustration) in **Appendix C**.

We sincerely hope this work can bring some insights into the field of image outpainting. Thanks again to all reviewers for their valuable time to help improve our work.

---

> ### Comment · Area_Chair_jkGw · 2023-11-15
>
> Acknowledged we have received your response. We will take those into consideration.
>
> Thanks,
> AC

---

### Meta-Review · Area_Chair_jkGw · 2023-12-09

**Metareview:**

This paper presents PQDiff, an approach for image outpainting that generalizes to arbitrary and continuous ratios, in a single step. The key to this capability is a relative positional embedding, which encodes different positional relationships. Experimental results validate the effectiveness of the proposed model.

Strengths:

Most of the reviewers, including me, agree on the following merits:
* The paper presents a simple yet effective method. It ensures continuous and flexible ratios in the outpainted images and generating the outputs in a single pass.
* The paper resolves a practical problem, and is well written.
* Most experimental evidences are convincing.
Along with them, several reviewers also comments:
* The approach is generalizable to the GAN-based models, and
* Very nice supplemental material.

Weaknesses:

The weaknesses vary between individual reviewers. Reviewer urTZ requested more justifications on continuous multiples and the quantitative results for outpainting in an arbitrary position; Reviewer xgEg concerned on the description, e.g., in unclearly justifying the need for multiple runs in GAN or MAE-based methods; inadequately explaining the advantages of its own method, and ambiguities in its model's design and implementation, as highlighted in its figures, equations, and treatment of image dimensions and positional relationships.

**Justification For Why Not Higher Score:**

Having reviewed the revised paper, along with all comments and discussions, I recommend accepting it for a poster presentation. While I acknowledge the paper's strengths, there is still potential for enhancement in the presentation of figures, articulation of claims, and detailing of descriptions.

**Justification For Why Not Lower Score:**

The authors have addressed the majority of the clarity concerns raised by the reviewers, and all reviewers have responded to the rebuttal, confirming their satisfaction with the explanations provided. The paper has been improved accordingly. Therefore, I find no grounds to justify a rejection.

---

### Decision · Program_Chairs · 2024-01-16

Accept (poster)